# Description of a New Cobra (*Naja* Laurenti, 1768; Squamata, Elapidae) from China with Designation of a Neotype for *Naja atra*

**DOI:** 10.3390/ani12243481

**Published:** 2022-12-09

**Authors:** Sheng-Chao Shi, Gernot Vogel, Li Ding, Ding-Qi Rao, Shuo Liu, Liang Zhang, Zheng-Jun Wu, Ze-Ning Chen

**Affiliations:** 1Key Laboratory of the Ministry of Education for Ecology of Rare and Endangered Species and Environmental Protection, Guangxi Normal University, Guilin 541004, China; 2Guangxi Key Laboratory of Rare and Endangered Animal Ecology, Guangxi Normal University, Guilin 541004, China; 3CAS Key Laboratory of Mountain Ecological Restoration and Bioresource Utilization & Ecological Restoration Biodiversity Conservation Key Laboratory of Sichuan Province, Chengdu Institute of Biology, Chinese Academy of Sciences, Chengdu 610041, China; 4Society for Southeast Asian Herpetology, D-69115 Heidelberg, Germany; 5Kunming Institute of Zoology, Chinese Academy of Sciences, Kunming 650201, China; 6Kunming Natural History Museum of Zoology, Kunming Institute of Zoology, Chinese Academy of Sciences, Kunming 650223, China; 7Guangdong Key Laboratory of Animal Conservation and Resource Utilization, Guangdong Public Laboratory of Wild Animal Conservation and Utilization, Institute of Zoology, Guangdong Academy of Sciences, Guangzhou 510260, China

**Keywords:** Asian cobras, *Naja kaouthia*, *Naja atra*, taxonomy

## Abstract

**Simple Summary:**

Cobras (*Naja* Laurenti, 1768) are a group of well-known highly venomous snakes, which cause numerous cases of snakebites every year, especially in South Asia and Southern China. Taxonomic framework is essential for the medical treatment of snake bites and accurate antivenin development. However, the taxonomy of Asian cobras is still puzzling, especially for the widespread species Monocled Cobra (*N. kaouthia* Lesson, 1831). This study provided new materials and understanding for the taxonomy of this species by combining mitochondrial phylogenetic analysis and morphological comparisons based on samples from a vast area in Asia. The results showed that the Chinese population of *N. kaouthia* represents a new species. This study also provided new data for *N. atra* and designated a neotype for it. Furthermore, the subspecies *N. naja polyocellata* was resurrected and recognized as a full species, *N. polyocellata*
**comb. nov.**, and the subspecies *N. sumatrana miolepis* was also resurrected. This study highlighted the necessity to evaluate the effectiveness of cobra antivenin based on a comprehensive taxonomic framework.

**Abstract:**

Taxonomic frameworks for medically important species such as cobras (genus *Naja* Laurenti, 1768; Squamata, Elapidae) are essential for the medical treatment of snake bites and accurate antivenin development. In this paper, we described the former *N. kaouthia* populations recorded from China as a new species and designated a neotype for *N. atra*-based morphological and mitochondrial phylogenetic analysis. The new species *N*. *fuxi*
**sp. nov.** was morphologically diagnosed from *N. kaouthia* by (1) regular single narrow crossband present on the middle and posterior parts of the dorsum (3–15, 7.9 ± 2.7, *n* = 32) and the dorsal surface of the tail (1–6, 4.2 ± 1.1, *n* = 32) of both adults and juveniles, buff-colored with dark fringes on both edges, vs. South Asian populations (*n* = 39) and Southeast Asian populations (*n* = 35) without cross bands, with irregular cross bands or multiple light-colored crossbands pairs, or densely woven lines; (2) small scales between the posterior chin shields, usually three (40%) or two (37%), rarely four (13%), or one (10%) (*n* = 30) vs. mostly one (81%) and rarely two (19%) (*n* = 28); (3) ventrals 179–205 (195.4 ± 6.7, *n* = 33) vs. South Asian populations 179–199 (188.7 ± 5.9, *n* = 12); Southeast Asian populations 168–186 (177.8 ± 4.9, *n* = 18). Phylogenetically, the new species forms an independent sister clade to the clade including *N. atra*, *N. kaouthia*, *N. oxiana* and *N. sagittifera*. Furthermore, the subspecies *N. naja polyocellata* should be resurrected and recognized as a full species, *N. polyocellata*
**comb. nov.**, and the subspecies *N. sumatrana miolepis* should be resurrected.

## 1. Introduction

Cobras of the genus *Naja* Laurenti, 1768 are a group of well-known highly venomous snakes that cause numerous cases of envenomation every year over an enormous geological area, especially in South Asia, southern China [1,2] and Southeast Asia. Under the current taxonomy framework, the composition of cobra toxins differs within species from different geographical populations, which could lead to antivenin deficiencies [3,4,5,6]. Recent studies on elapid species have shown that widespread medically important species may still require taxonomic revisions [5,7]. It seems necessary to examine the current taxonomy for those species with geographical variation in venom toxins.

Asiatic cobras have a complex taxonomic history (e.g., [8,9,10,11,12,13,14,15,16,17,18]). All Asiatic cobra populations were previously considered as a single species, *Naja naja* (Linnaeus, 1758) (in older references also referred to as *Naja tripudians*) with multiple subspecies and color variants [19,20]. Currently, twelve Asian species are regarded as valid (Figure 1): *N. atra* Cantor, 1842; *N. arabica* Scortecci 1932; *N. kaouthia* Lesson 1831; *N. mandalayensis* Slowinski and Wüster, 2000; *N. naja* (Linnaeus, 1758); *N. oxiana* (Eichwald, 1831); *N. philippinensis* Taylor, 1922; *N. sagittifera* Wall, 1913; *N. samarensis* Peters, 1861; N. *siamensis* Laurenti, 1768; *N. sputatrix* Boie, 1827 and *N. sumatrana* Müller, 1887 [21,22,23]. These species all belong to the subgenus *Naja* Laurenti, 1768 except *N. arabica* Scortecci, 1932, which is assigned to the subgenus *Uraeus* Wagler, 1830 [24].

The Monocled Cobra, *Naja kaouthia*, is widely recorded in northeastern India, Bangladesh, Bhutan, Nepal, Myanmar, Cambodia, Laos, Northern Malaysia, Southern Vietnam, Thailand and southern China [25] and was firstly described from Bengal by Lesson (1831). However, recent research implied that considerable diversification has occurred among different geographical populations (China, Thailand and Bangladesh) of this species [5,26,27,28,29]. Multiple variations among populations of *N. kaouthia* regarding coloration, spitting behavior and especially venom toxin composition were found; the venom of the Yunnan population of *N. kaouthia* lacks α-cobratoxin while that of the Thailand population has it [3,4,5,6]. These findings suggest that currently *N. kaouthia* may include multiple species and multiple populations, especially the Yunnan Population, necessitating further taxonomic research.

Cobras recorded from China were once regarded as one species with two subspecies under the name *Naja naja* [30] but later became regarded as two species, namely *N. atra* and *N. kaouthia* [31]. The *N. kaouthia* in China was recorded from southwestern China, including South and Southwestern part of Yunnan, Southwestern part of Sichuan, Xizang (Tibet Autonomous Region), and Guangxi [30,31,32]. The *N. atra* in China was recorded from southern China, south of the Yangtze River, including Zhejiang, Fujian, Taiwan, Guangdong, Hainan, Guangxi, Macao, Hongkong, Jiangxi, Anhui, Hubei, Hunan and Guizhou [30,31]. A comprehensive study on *N. atra* (including samples from multiple sites in southern China and northern Vietnam, covering nearly all distribution areas) has shown that the lineage from Yunnan Province, China is deeply divergent from samples from other localities [33]. However, the type locality of *N. atra* is Zhoushan Island, Zhejiang Province, eastern China. This leads to a taxonomic puzzle about the status of cobras from Yunnan.

To further understand these problems, we compiled a data set including a series of fresh cobra samples from China together with available cobra sequences and reconstructed a comprehensive mitochondrial phylogenetic tree. Together with morphological data based on 143 specimens and photos of 208 living individuals of nine taxa from Asia, we found the populations of *Naja kaouthia* from southwestern China represent a new species that we described herein. We also designated a neotype for *N. atra* for future taxonomic study on the genus.

## 2. Materials and Methods

### 2.1. Sampling

In this study, a total of 14 “*Naja kaouthia*” specimens and 6 *N. atra* specimens from China were collected (Table 1). Specimens were euthanized and then fixed in 80% ethanol before deposition in the Herpetology Museum of Chengdu Institute of Biology, Chinese Academy of Sciences, Chengdu City, Sichuan Province, China (CIB). Other abbreviations for institutions are as follows: Kunming Institution of Zoology, Chinese Academy of Sciences, Kunming, China (KIZ); Naturhistorisches Museum Wien, Vienna, Austria (NMW), Zoologisches Museum Hamburg, Hamburg, Germany (ZMH); Guangxi Normal University, Guilin, China (GXNU); Zoologisches Forschungsmuseum (formerly Forschunginstitut und Museum) “Alexander Koenig,” Bonn, North Rhine-Westphalia, Germany (ZFMK); Naturhistorisches Museum Basel, Basel, Switzerland (NMBA); Museum für Naturkunde (formerly Zoologischen Museum), Leibniz-Institut für Evolutions-und Biodiversitätsforschung, Universität-Humboldt zu Berlin, Berlin, Germany (ZMB).

### 2.2. Molecular Phylogenetic Analysis

Genomic DNA was extracted from muscle or liver tissues using QIAamp DNA Mini Kit (QIAGEN, Hilden, Germany). We sequenced three mitochondrial genes for analysis: cytochrome *b* (cyt *b*) (primers cited from Burbrink et al. [34]), NADH dehydrogenase subunit 4 (ND4) (primers cited from Arevalo et al. [35]) and cytochrome C oxidase 1 (COI) (primers cited from Che et al. [36]). PCR amplifications were performed in 25 μL reactions (12.5 μL I-5 2×High-Fidelity Master Mix, 10 μL ddH_2_O, 1 μL F-primers, 1 μL R-primers, 0.5 μL DNA template) under the following cycling conditions: initial denaturation for 2 min at 95 °C, 35 cycles with denaturation at 94 °C for 40 s, annealing at different temperatures (48.5 °C for cyt *b* and COI, 56 °C for ND4) for 25 s, extension at 72 °C for 15 s, and final extension for 2 min at 72 °C. PCR products were sequenced by Beijing Qingke New Industry Biotechnology Co., Ltd., Beijing, China. Raw trace files for sequences were edited in Genious 7 (Biomatters Limited, Auckland, New Zealand) before constructing alignments using MEGA 7 ([37]). Due to differences in taxon sampling for each gene, we reconstructed separate alignments for phylogenetic analysis. The first one was based on a concatenated sequence alignment using cyt *b* and ND4, while the other was based solely on COI. Sequences were uploaded to GenBank (accession numbers see in Table 1). Available cyt *b*, ND4 and COI sequences of all 12 recognized Asian cobra species and 20 of 21 recognized African cobra species (except *Naja nana* [38]) were downloaded from GenBank (Table 1). Our DNA dataset employed sequences of 86 specimens from GenBank and 20 specimens collected in this study. *Ophiophagus hannah* was chosen as an outgroup according to Li [39]. Optimal models of sequence evolution of nucleotide substitution were identified by Bayesian Information Criterion (BIC) using Partition finder 2.1.1 [40]. We performed maximum likelihood (ML) analysis using RaxML v8 [41]. The first ML analysis was implemented in RaxML v8 [41] following the GTRGAMMA model with 1000 fast bootstrap replicates to assess node support. We considered bootstrap proportions of 70% or greater to be strong support for existence of a clade following [42]. The best evolution models are shown in Appendix A. Bayesian inference phylogenetic trees were inferred using MrBayes 3.2 [43]. We used a random starting tree and four independent runs with a maximum of 20 million generations each, sampled every 1000. Runs were stopped when the average standard deviation of split frequencies had reached 0.001. The first 25% of each run were discarded as burn-in. Nodes with Bayesian posterior probabilities (BPP) of 0.95 and above were considered well supported [44]. Genetic distance between species with uncorrected p-distance model was estimated using MEGA 7.

### 2.3. Morphological Analysis

Measurements of head and head scales were taken with digital calipers and rounded to the nearest 0.1 mm; snout–vent length and tail length were taken with a measuring tape and rounded to the nearest 1 mm. Terminology and descriptions follow Slowinski (1994), Vogel et al., (1994) and Kuch et al. (2005) [45,46,47]. Morphometric and meristic characters are abbreviated as follows: total length (TL), from the tip of snout to the tip of tail; snout–vent length (SVL), from the tip of snout to anterior margin of cloaca; tail length (TaL), from posterior margin of cloaca to the tip of tail; ratio of tail length to total length (TaL/TL); head length (HL), from snout tip to the end of parietals suture; head width (HW) was measured at the widest part of the head on posterior side; head height (HH), at the maximal highest part of the head; interorbital distance (IOD); the dorsal scale rows (DSR) were counted at one head length behind the head, at midbody, and at one head length before the vent; ventral scales (VEN) were counted according to Dowling [48], half ventrals were counted as one. The enlarged shield(s) anterior to the first ventral were regarded as preventral(s); for subcaudals (SC), the first scale under the tail meeting its opposite was regarded as the first subcaudal scale, and the unpaired terminal scute was not included in the number of subcaudals; paired scales on the head were counted on both sides of the head and presented in left/right order; supralabials (SL); infralabials (IL) were considered scales and shields that are completely below a supralabial and border the gap between lips. The number of crossbands on the body (BB) and on the tail (BT) were counted. Sex was determined by making a small incision caudal to the vent to visually inspect for the existence of hemipenes. Hemipenis terminology follows Dowling (1951), and the organs were prepared based on Jiang [49]. Fangs adjusted for spitting or not were judged according to Wüster and Thorpe [17] and Young et al. [50]. For morphological comparison, a total of 143 specimens from 12 taxa were examined and photos of 208 living cobras from Asia from institutional and personal collections and the Global Biodiversity Information Facility (GBIF) were analyzed (Appendix A). Morphological data for some congeners of the subgenus *Naja* were obtained from the literature (Appendix A). The statistical summaries of scale counts are presented as ranges (Mean ± Standard deviation, Sample size).

## 3. Results

### 3.1. Molecular Results

The concatenated alignment for cyt *b* and ND4 was 1687 bp in length (1028 + 659 bp, respectively) and contained a total of 33 taxa of *Naja*. The phylogenetic trees resulted in essentially identical topologies by BI and ML analysis on cyt *b* + ND4 (Figure 2). The resulting phylogeny showed a topology similar to that of recent studies [27,51]. *N. arabica* formed a clade (K; bootstrap proportions/BPP 100/1.00) with the African cobra subgenus *Uraeus*, which is similar to the phylogeny in Trape et al. (2009) [52]. All other Asian cobras (subgenus *Naja*) formed a large sister clade (clade H; 97/1.00) to the subgenus *Uraeus*. These Asian cobras were divided into three smaller clades: the basal *N. naja* clade (J; 100/1.00), the Southeastern Asian clade (I; 100/1.00) ((((*N. sumatrana* + *N. siamensis*) + *N. mandalayensis*,) + (*N. samarensis* + *N. philippinensis*)) + *N. sputatrix*), and the Pan-Asian clade (E; 69/0.91) (the “*N. kaouthia*” from Southwestern China + (((Southeastern Asian *N*. *kaouthia* + *N. sagittifera*) + *N. oxiana*) + *N. atra*)) based on cyt *b* + ND4. The samples of *Naja atra* from southern and southeastern China clustered in a clade (D; 100/1.00), and further formed a larger clade (C; 90/1.00) with southeastern Asian *N*. *kaouthia*, *N. sagittifera*, and *N. oxiana*. The sample under the name *N. atra* from Yunnan, Southwestern China clustered with “*N. kaouthia*” from Sichuan and Yunnan, Southwestern China, and formed an independent clade (F; 100/1.00) in the Pan-Asian clade. The samples of southeastern Asia *N. kaouthia* from Myanmar to Vietnam formed a clade (A; 98/1.00) sister to insular species *N. sagittifera*.

The COI alignment was 627 bp in length and contained ten taxa of *Naja*. The BI and ML analysis resulted in essentially identical topologies (Figure 3). The *N. kaouthia* from South Asia (India and Bangladesh) formed a clade (L; 99/1.00), which formed a larger clade (M; 57/0.94) with Southeastern Asian clade (A; 98/0.98). The “*N. kaouthia*” samples from Southwestern China formed a clade paraphyletic to the south and Southeastern *N. kaouthia* clade (M) and *N. atra* clade (D).

From the aspect of genetic distance, considerable divergence existed between different clades under the name *N. kaouthia*. The Southwestern China clade (F) was divergent from the South Asian clade (L, 2.0–3.0% for COI) and Southeast Asian clade (A) (2.5–2.9% for COI, 5.3–6.7% for cyt *b*). These are comparable to the genetic distance between of *N. kaouthia* Southwestern China clade and *N. atra* (2.5–3.2% for COI; 4.1–5.0% for cyt *b*), *N. kaouthia* South Asian clade and *N. atra* (2.7–3.4% for COI, 4.6–6.3% for cyt *b*) (Table 2 and Table 3). The latter two species were long regarded as separated species [25,31,53,54]. The genetic distances between the Southwestern China clade and Southeastern Asian clade of *N. kaouthia* are also comparable for some other known species pairs from Asia: *N. siamensis* and *N. sumatrana* (4.3–5.0% for cyt *b*), *N. siamensis* and *N. mandalayensis* (3.2–3.9% for cyt *b*), *N. sumatrana* and *N. mandalayensis* (3.2–3.9% for cyt *b*). Considerable divergence also existed between the South Asian clade (L) and Southeast Asian clade (A) of *N. kaouthia* (1.8–2.4% for COI).

For genetic divergence within a clade, samples of “*N. kaouthia*” with large geographical distance had a small divergence. The samples of the Southeast Asian clade had minor genetic divergences (0.0% to 1.2% for cyt *b*) between geographical populations from Southern Myanmar, Southern Thailand, and Vietnam, whose largest geographical distance is more than 1300 km. In contrast, the insular species *N. sagittifera*, which originated in the Andaman Islands, had a larger genetic divergence (2.4–2.5% for cyt *b*) with the *N. kaouthia* Southeast Asian clade, which is about 300 km away. The genetic divergence between samples of the *N. kaouthia* Southwestern China clade were minor (0.0–0.3% for cyt *b*, 0.0–0.2 for COI) while they are maximally separated by about 500 km. The genetic distances between the samples of *N. kaouthia* South Asia clade (Rangpur, Bangladesh and Mizoram, India; geographical distance about 400 km) were 0.0–0.6 for COI. These minor genetic distances within clades of *N. kaouthia* were distinctly smaller than those between the Southwestern China clade and Southeastern Asian clade, which further indicated that the latter is interspecific.

### 3.2. Morphological Results and Taxonomic Conclusion

Comparisons of selected morphological characters and spitting behavior for the subgenus *Naja* are listed in Appendix A.

The “*Naja kaouthia*” from Sichuan and Yunnan, Southwestern China is morphologically different from the *N. kaouthia* from South Asian and Southeast Asian populations by (1) regular single narrow crossband present on the middle and posterior parts of the dorsum (3–15, 7.9 ± 2.7, *n* = 32) and dorsal surface of tail (1–6, 4.2 ± 1.1, *n* = 32) of both adults and juveniles, buff-colored with dark fringes on both edges, vs. South Asian populations (*n* = 39) and Southeast Asian populations (*n* = 35) without cross bands or with irregular cross bands or multiple light-colored cross bands pairs or densely woven lines; (2) small scales between posterior chin shields were usually three (40%) or two (37%), rarely four (13%), or one (10%, *n* = 30) vs. one for the South Asian clade (*n* = 8) and mostly one (72%) and rarely two (28%) for the Southeastern Asian clade (*n* = 18); (3) ventrals 179–205 (195.4 ± 6.7, *n* = 33) vs. the South Asian clade 179–199 (188.7 ± 5.9, *n* = 12) and Southeast Asian clade 168–186 (177.8 ± 4.9, *n* = 18).

In conclusion, the samples of “*Naja kaouthia*” from Southwestern China represent a new species, which is phylogenetically and morphologically different from *N. kaouthia* from South Asia and Southeastern Asia.

### 3.3. Taxonomic Account

#### 3.3.1. *Naja atra* Cantor, 1842

Figure 4A, Figure 5 and Figure 6A,B

[English name: Chinese Cobra]

[Chinese name: 舟山眼镜蛇]


**Synonyms:**


Naja tripudians nigra Gray, 1834

*Naja tripudians* var. scopinucha Cope 1859

*Naja tripudians* var. *larrata* Steindachner 1867 (nomen praeoccupatum)

*Naja tripudians* var. *unicolor* W.C.H. Peters in Martens, 1876 (nomen substitutum)

*Naia tripudians* var. *fasciata*–Boulenger 1896 (populations of part b from Kiu Kiang, Canton and Hainan)

**Remark**. *Naja atra* Cantor, 1842 was described from Chusan (=Zhoushan 舟山, Zhejiang Province), China and no holotype has been designated [25,55]. According to Lin et al. [33], samples of *Naja atra* from vast areas across southern China (except two samples from Yunnan, which are allocated to the new species described in this paper) and Northern Vietnam form two clades with a small divergence (Vietnam + southern China + southwestern China; eastern + southeastern China) based on analysis of 12 microsatellite loci and 1117 bp of the mitochondrial cytochrome *b* gene. This supports the synonymisation of *Naja tripudians* var. *scopinucha* Cope, 1859 (type locality: Canton River), *Naja tripudians* var. *larrata* Steindachner, 1867 (type locality: Hongkong), *Naja tripudians* var. *unicolor* Peters, 1876 (substitute name), *Naia tripudians* var. *fasciata* Boulenger, 1896 (part b) (Kiukiang, Canton, Hainan) with *Naja atra* Cantor, 1842. To avoid future confusion with other Asiatic *Naja* species, we designate a neotype from the type locality Zhoushan Island, Zhejiang, China, and redescribe this species based on examined specimens from Southern China. The identification of these examined specimens of *N. atra* were further confirmed by their localities at or near the sampling localities of clade A & B of Lin et al. [33]. Descriptions are based on 30 specimens examined from Southern China and photos of six unvouchered living individuals from Fujian and Guangdong provinces, China (Appendix A).

**Neotype**. CIB 12273, subadult female, collected from Zhoushan Island, Zhejiang Province, China, in July 1983.

**Diagnoses**. This species is characterized by the following combination of characters: (1) body scales smooth, scale rows at anterior body 20–27 (23.7 ± 1.7, *n* = 29), mid-body 19–23 (20.8 ± 1.0, *n* = 31), posterior body 15 (*n* = 29); (2) ventrals 161–175 (170.9 ± 3.2), subcaudals 33–50 (45.0 ± 3.7) (*n* = 29); (3) total length for adults 1041–1412 mm (1215.1 ± 117.6 mm), tail length 130–222 mm (176.9 ± 25.6 mm) (*n* = 13), tail length/total length 0.119–0.164 (0.146 ± 0.012, *n* = 19, including adults and juveniles); (4) usually one small scale between posterior chin shields (72%) rarely two (28%) (*n* = 29); (5) dorsum mostly dark or deep brown in adults and juveniles, few specimens are light brown; (6) throat pale usually without dark mottling, distinct ventrolateral throat spots, followed by a broad dark or deep brown band; (7) hood markings mostly formed as a pale heart-shaped with a dark center and two small dark dots on the side; (8) most adults and juveniles with narrow light-colored crossbands on dorsal middle and posterior body 5–21 (11.3 ± 3.7, *n* = 27) and dorsal tail 1–7 (4.6 ± 1.8, *n* = 26); (9) hemipenial spines on distal part weakly connected at base, arranged in rows; and (10) fangs not modified for spitting, venom discharge orifice large; however one case of spitting was observed in captivity [56].

**Description of Neotype**. (Figure 4A) Head moderate, triangular, widest at middle parietals. Body dimensions: snout-vent length 730 mm, tail length 122 mm, head length 20.2 mm (snout to end of parietal suture), head width 19.8 mm, head height 8.8 mm, distance between eyes 10.3 mm.

**Body scalation**. Ventrals 175, subcaudals 45 pairs. Dorsal scales smooth, rows 22-21-15; elongated, gradually broader toward ventrals; vertebral scales similar with adjacent dorsal scales.

**Head scalation**. Scales smooth. Rostral slightly visible in dorsal view. Nasals large, divided, nostril vertically oval, larger than half the eye, surrounded by prenasal anteriorly and postnasal for the remainder of the sides. Loreal scale absent. Preocular 1/1 (left/right), bordering widely with third supralabial, supraocular, prefrontal, and slightly with eye, internasals and postnasals. Internasal paired, widely in contact with each other. Prefrontal paired, in shape of right trapezoid, in broad contact with one another. Frontal moderate, longer than wide, shield-shaped. Parietals large and paired, largest length about 1.4 times of frontal length, each bordering frontal, upper two supraoculars, anterior and posterior temporals, and two small scales on dorsal head. Eyes rounded, moderate, vertical diameter equal to height of border of fourth supralabial. Postoculars 3/3, middle postocular larger than the two other ones, much smaller than preocular, the lowest widely in contact with fourth and fifth supralabials. Temporals 2+3/2+2; lower anterior temporals hexagonal, longer than high; lowest posterior temporal is largest on lateral head, shape irregular. Supralabials 7/7; first two smallest, bordering nasals; third and fourth higher than wide, bordering eye; the seventh longest. Mental triangular in front and ventral view of head. Infralabials 9/9; first to third in contact with anterior chin shields; the first pair contacting each other, and anterior edge of anterior chin shields; the fourth in contact with both anterior and posterior chin shields; the fifth small and poorly developed, barely seen in ventral view of head, leaving the fourth in broad contact with the sixth; the sixth in contact with posterior chin shields; the seventh to ninth elongated, not contacting chin shields. The two pairs of chin shields butterfly shaped. Only one elongated scale separating posterior chin shield.

**Coloration in preservative**. Dorsal and upper lateral head brown, lower head paler, sutures of the first to sixth supralabials and fourth to seventh supralabials partially edged with narrow deep brown borders. Dorsum and tail deep brown, 16 light brown crossbands present on body, covering one row of dorsal scales, with narrow dark fringes, crossbands indistinct anteriorly, gradually distinct on posterior body and tail. Seven light brown crossbands present on dorsal tail, with clear dark fringes. Most crossbands on body and tail bifurcate near ventrals, with a dark spot at the base of crossbands. Hood markings mostly pale brown, heart-shaped; with a distinct larger dark core patch and two smaller dark side patches, broadly in contact with pale brown edges of the side of the neck. A small dark patch present on borders of eighth ventral scale. Dorsal scales between hood marking and small dark patch pale brown. Ventral surfaces of head and neck mostly white, stained with light yellow; first broad band on ventrals dark, commencing at 14th ventral and covering almost five ventrals; three ventrals (19th–21st) after first dark band almost yellowish-cream, then becoming gradually brown, completely deep brown by the 24th ventral; ventrals after 24th mostly uniformly deep brown, a few with pale lateral edges due to light brown crossbands on dorsum partially intruding upon ventrals; cloacal shield and subcaudals light brown, paler at places where light brown crossbands on dorsal tail intrude.

**Fang.** Fangs firm, short, tip reaching middle of fourth supralabial; not modified for spitting, venom discharge orifice relatively large and elongated.

**Hemipenes.** (Figure 6A,B) Description based on five adult males (Appendix A). Hemipenis bilobed, slender, covered with dense small spines, smaller and sparser at proximal part, larger and denser on distal part; spines on distal part weakly connected at base, arranged in rows. Two pairs of distinctly developed lateral folds present on the trunk from sulcate view, nearly identical, highly raised and covered with dense spines. Lower lobes located at two-thirds of the trunk, near the base; upper lobes raise at two-thirds of the trunk and reach the first third; a transverse constriction divides the lobes from asulcate view. Sulcus forked, centripetally spiral to the top at asulcate side, tip of sulcus not visible from sulcate view. Sulcus lip well developed and raised, covered with dense spines. Hemipenes reaching the 10th–12th subcaudal when not everted, forked at 6th–9th subcaudal; hemipenial retractor muscle reaching 29th–30th subcaudal, forked at 11th–14th subcaudal. Bifurcation ratio 59–67%, hemipenes length/snout-vent length 3.5–4.6%.

**Variation.** (Figure 5) Body length for adults 891–1194 (894 ± 181) mm, total length 1041–1412 (1215 ± 118) mm (*n* = 13). The coloration on the body is mostly dark or deep brown on adults as well as juveniles or hatchlings. Crossbands on body are regular on most individuals (64%, *n* = 36) while crossbands on a small part of population are irregular (some forked) (31%), or absent (6%). Hood markings mostly heart-shaped (79%, *n* = 33) with two small dark dots (82%, *n* = 33); mostly one (72%, *n* = 29), rarely two (28%), small scales between posterior chin shields. The first ventral crossband begins between at the level of 12th–17th (14.2 ± 1.1, *n* = 33) ventrals.

**Distribution.** Central and eastern South China (areas south of the Yangtze River: Zhejiang, Fujian, Southern Anhui, Jiangxi, Hunan, Western Hubei, Guizhou, Guangdong, Guangxi, Hainan, Taiwan), northern Vietnam.

#### 3.3.2. *Naja kaouthia* Lesson, 1831

Figure 3A, Figure 4A and Figure 5

[English name: Monocled Cobra]

[Chinese name: 孟加拉眼镜蛇]

*Naja kaouthia* Lesson, 1831. Holotype: unknown (*fide* Leviton et al., 2003)


**Synonyms:**


*Naja tripudians* var. *fasciata* Gray 1830

*Naja larvata* Cantor, 1839

*Naga tripudians monocellata* Nicholson, 1874

*Naja tripudians viridis* Wall, 1913a

*Naja kaothia*–Wall, 1913b (nomen incorrectum)

*Naja naja kauothia*–Kabara and Fischer, 1972 (nomen incorrectum)

*Naja kauthia*–Khole, 1991 (nomen incorrectum)

*Naja kauthia suphanensis* Nutaphand, 1986

*Naja naja combodia* Khole, 1991 (lapsus calami)

**Remarks.** Present samples of *Naja kaouthia* from the South Asian population and Southeast Asian population form sister clades. Since the type locality of *N. kaouthia* is the Bengal region [57], those specimens from northeastern India, Bangladesh and adjacent areas including Bhutan, Nepal, and southern Tibet Autonomous Region, China should be treated as *N. kaouthia sensu stricto* [5,26,27,28,29]. The following diagnoses of *N. kaouthia* are based on data from South Asia, including data of 13 preserved specimens and 32 living individuals. Measurements and scale counts of three adult females and one adult male were cited from Lalremsanga et al. [58]. Data of living individuals were based on photos of from literatures [28,32,59,60], personal collections of Vogel Gernot and Vivek R Sharma, and GBIF.com) (Appendix A).

**Diagnoses:** (1) body scales smooth, scale rows at anterior body 22–29 (26.3 ± 2.2, *n* = 12), mid-body 19–28 (21.4 ± 2.3, *n* = 13), posterior body 13–20 (14.8 ± 1.8, *n* = 12); (2) ventrals 179–199 (188.7 ± 5.9, *n* = 12), subcaudals 46–57 (52.8 ± 2.9, *n* = 13); (3) total length for adults 937–1712 mm (1266.4 ± 248.1 mm, *n* = 10), tail length 137–225 mm (174.1 ± 30.0 mm, *n* = 10), tail length/total length 0.126–0.159 (0.139 ± 0.009, *n* = 11, including adults and juveniles); (4) small scale between posterior chin shields mostly one (81%), rarely two (19%, *n* = 28); (5) dorsum of adults mostly brown, juveniles olive-brown, yellowish-brown, or deeper; (6) dorsal middle and posterior body and dorsal tail without crossbands or with irregular cross bands or multiple light-colored cross bands pairs or densely woven lines; (7) throat pale without dark mottling, ventrolateral throat spots usually distinct, usually followed by a broad dark band, the band occasionally light brown; (8) hood pattern usually a monocellate light brown or white circle, often with a large deep colored center and two lateral dots; in few cases, monocellate pattern absent; (9) populations from lowlands of Bangladesh and adjacent area with multiple light-colored crossbands pairs or dense woven lines on body, or few with one or two clear crossbands on neck after hood, populations from mountainous area in southern slope of Himalayan without crossbands on body except few with one or two crossbands after hood, populations from Thailand and peninsular Malaysia light brown, populations from southern Vietnam deep brown; (10) spitting venom.

**Fangs.** (Description based on examination of one adult female CIB 12285 from southern Tibet, China) Firm and short, not exceeding third supralabial; not modified for spitting, venom discharge orifice relatively large and elongated.

**Variation.** (Figure 7) The coloration of South Asian clade *Naja kaouthia* varies between geographical regions. The populations from the lowlands (Odisha, West Bengal and Tripura of India; Bangladesh) (Figure 7A–E) are yellowish-brown or deep brown dorsally, usually with distinct multiple dense light-colored crossbands, or at least a distinct broad crossband on the neck behind the hood pattern; only one out of 16 individuals uniformly brown dorsally. The populations from southern slope of Himalayan Mountains (Mizoram, Assam, and Sikkim of India; Bhutan, Nepal) uniformly brown in adults and mostly olive-brown or deep brown in juveniles. Hood pattern usually a monocled light brown or white circle, often with a large deep colored center and two lateral dots; in few cases, monocled pattern absent (Figure 7F) or connected with light colored patches on lateral neck (Figure 7G). Adults and juveniles from lowlands in West Bengal and adjacent areas without multiple light-colored crossbands on body or solely with one or two clear crossbands on neck after hood markings, only small part of populations from mountainous area along southern slopes of Himalayas without multiple crossbands on body.

The Southeastern Asian clade is also morphologically different across regions in coloration. Adults from Thailand and peninsular Malaysia are light brown, the population from southern Vietnam is deep brown; juveniles usually darker, blackish brown, olive brown, or light brown (Figure 8). Ventrals of South Asian population 168–186 (177.8 ± 4.9, *n* = 18); subcaudals 49–58 (54.7 ± 2.4, *n* = 18); scale rows at anterior body 25–31 (27.0 ± 1.9, *n* = 20), mid-body 20–23 (21.2 ± 0.7, *n* = 19), posterior body 15–17 (15.4 ± 0.8, *n* = 19); venom spitting behavior was not observed [17,50].

**Distribution.** South Asian clade: Northeastern India, Nepal, Bhutan, southern Xi-zang Autonomous Region (Tibet) of China, Bangladesh. Southeastern Asian clade: Southern Myanmar, central and southern Thailand, Cambodia, central and southern Laos, southern Vietnam, Malay Peninsula.

#### 3.3.3. *Naja fuxi*
**sp. nov.** Shi, Vogel, Chen, Ding

Figure 4C, Figure 6C,D, Figure 9 and Figure 10.

ZooBank LSID: E13EEB16-7256-4589-8D22-F708B2692D86


**Chresonyms:**


*Naja naja kaouthia*—Zhao et al., 1998; Yang and Rao, 2008

*Naja kaouthia*—Zhao, 2003

*Naja atra*—Lin et al., 2012 (Clade C)

*Naja kaouthia*—Huang, 2021

**Holotype.** Adult male CIB DL2018053147 (Figure 4D), collected from Tongde Town, Renhe District, Panzhihua City, Sichuan Province, China (26.710278° N, 101.564167° E, 1400 m a.s.l.) on 31 May 2018, by Ze-Ning Chen and Li Ding. The holotype is a roadkill but was only barely damaged.

**Paratypes.** (33 specimens) One adult female CIB DL2018081005 collected from Renhe District, Panzhihua, Sichuian Province, China (26.466293° N, 101.740758° E, 1248 m a.s.l.) on 10 August 2018, by Li Ding. One adult female CIB 101318; four adult males CIB 012303, CIB 012295, CIB 012288, CIB 012289; four juvenile females CIB 012290–012292, CIB 012294; and one juvenile male CIB 012293 were collected from Miyi Cunty, Panzhihua City, Sichuan Province in 1985. One adult male CIB 098874 and adult female CIB 098875 were collected from Longhua village, Caocang Town, Miyi Conty, Panzhihua City, Sichuan Province, China (26.5628° N, 102.0402° E, 1373 m a.s.l.) by Yue-Ying Chen and Bo Cai on 19 December 2012. One unknown sex juvenile CIB DL0096 from Yanbian, Panzhihua, Sichuan, China. Six juveniles from Jiangcheng, Pu’er, Yunnan, China, including two females (CIB DL000070, CIB DL YNJC0068), three males (CIB DL000022, CIB DL000239, CIB DL000020) and one unknown sex (CIB DL R428). Two adults (male CIB 012296, female CIB 012297) and three female juveniles (CIB 012298–CIB 012300) from Lushui, Nujiang, Yunnan, China. Two adults from Yunnan, China (male CIB 012302, female CIB 012301) without accurate locality. One male juvenile KIZ 2020090301 from Menglian, Pu’er, Yunnan, China. One male juvenile KIZ 2020091201 from Ximeng, Pu’er, Yunnan, China. Three from Simao, Pu’er, Yunnan, China (male juvenile KIZ 20180801 and KIZ F20180066, male adult KIZ 090071). One male adult CIB 83796 from China without specific locality.

**Diagnoses.** (1) body scales smooth, scale rows at anterior body 19–29 (23.8 ± 3.1, *n* = 32), mid-body 19–27 (20.9 ± 1.5, *n* = 33), posterior body 12–19 (15.3 ± 2.9, *n* = 33); (2) ventrals 179–205 (195.4 ± 6.7, *n* = 33), subcaudals 45–61 (51.4 ± 9.7, *n* = 33); (3) total length in adults 690–1366 mm (1039.5 ± 207.6 mm, *n* = 15), tail length 110–201 mm (145.9 ± 27.8 mm, *n* = 15), tail length/total length 0.128–0.162 (0.141 ± 0.009, *n* = 32, including adults and juveniles); (4) small scales between posterior chin shields mostly three (40%) or two (37%), rarely four (13%) or one (10%) (*n* = 30); (5) dorsum light brown in adults, deep brown or black in juveniles (*n* = 32); (6) throat pale usually without dark mottling, ventrolateral throat spots distinct, followed by a broad light brown band; (7) hood markings usually a pale oval marking with narrow dark inner and outer border (73%), sometimes irregular residual patterns of a monocle (20%), rarely indistinguishable (7%), no dark side spots in any individual (*n* = 30); (8) both adults and juveniles with clear regular single narrow buff cross bands with dark fringes on middle, posterior dorsum 3–15 (7.9 ± 2.7, *n* = 32) and dorsal tail 1–6 (4.2 ± 1.1, *n* = 32); (9) hemipenial spines on distal part well connected at base, forming calyculate fold; (10) fangs not modified for spitting, venom discharge orifice relatively large.

**Comparisons.** The new species has long been identified as *Naja kaouthia*, but it is different from the latter by: (1) regular single narrow crossbands present on middle and posterior parts of the dorsum (3–15, 7.9 ± 2.7, *n* = 32), and dorsal surface of tail (1–6, 4.2 ± 1.1, *n* = 32) of both adults and juveniles, buff-colored with dark fringes on both edges vs. South Asian populations (*n* = 39) and Southeast Asian populations (*n* = 35) without crossbands or with irregular cross bands or multiple light-colored cross bands pairs or densely woven lines; (2) small scales between posterior chin shields usually three (40%) or two (37%), rarely four (13%) or one (10%) (*n* = 30) vs. mostly one (81%), rarely two (19%) (*n* = 28); (3) ventrals 179–205 (195.4 ± 6.7, *n* = 33) vs. South Asian populations 179–199 (188.7 ± 5.9, *n* = 12); Southeast Asian populations 168–186 (177.8 ± 4.9, *n* = 18).

The new species differs from *Naja atra* by: (1) ventrals 179–205 (195.4 ± 6.7, *n* = 33) vs. ventrals 161–175 (170.9 ± 3.2, *n* = 29); (2) small scales between posterior chin shields usually three (40%) or two (37%), rarely four (13%) or one (10%) (*n* = 30) vs. usually one (72%), rarely two (28%), *n* = 29; (3) dorsum coloration of adults light brown, crossbands on body and tail with dark fringes (*n* = 32) vs. usually dark, pale cross bands without recognizable fringes (*n* = 36); (4) hood markings mostly a pale oval marking with narrow dark inner and outer border (73%), some have an irregular residual pattern of a monocle (20%), others indistinguishable (7%), no dark side spots (*n* = 30) vs. mostly formed by a pale heart-shape (79%) with a dark center and two small lateral dark dots (*n* = 33); (5) hemipenial spines on distal part well connected at base, forming calyculate fold vs. spines on distal part weakly connected at base, arranged in rows.

From other cobras from South Asia, the new species differs as follows: from *Naja naja* by (1) hood markings usually form a complete monocle, in a few specimens incomplete, rarely absent vs. spectacle-shaped; (2) throat pattern distinct, lateral spots encroach on lowest dorsal scale row vs. throat pattern indistinct, lateral spots encroach on second dorsal scale row [14]; (3) regular single narrow crossbands present on middle and posterior parts of the dorsum (3–15, 7.9 ± 2.7, *n* = 32), and dorsal surface of tail (1–6, 4.2 ± 1.1, *n* = 32) of both adults and juveniles, buff-colored with dark fringes on both edges vs. mostly without regular light-colored crossbands on body (83%), some with brown and yellow densely woven lines on body (17%) (*n* = 46).

The new species differs from *Naja oxiana* by (1) fewer subcaudal scales 45–61 (51.4 ± 9.7, *n* = 33) vs. males 63–71 (68.0 ± 2.0); females 57–70 (63.3 ± 2.7); (2) hood markings usually as a distinct pale oval markings with narrow dark inner and outer borders vs. usually indistinct; (3) throat marking distinct, with lateral spots vs. throat marking indistinct, lateral spots absent; (4) crossbands on adults not present on neck vs. present [14].

The new species differs from *Naja sagittifera* by (1) more ventral scales 179–205 (195.4 ± 6.7, *n* = 33) vs. 175–176 (175.40 ± 0.55, *n* = ?) () for males, 183 for one female); (2) dorsum coloration light brown in adults, dark brown in juveniles, regular narrow crossbands with dark fringes on body and tail of adults as well as juveniles vs. juveniles dark with irregular or shark-fin-like outlines; (3) distribution confirmed in southwestern China and expected in adjacent Indochina vs. restricted to the Andaman Islands.

Differences from cobras from southeastern Asia: the new species differs from *Naja mandalayensis* by (1) hood markings usually a distinct pale oval marking with narrow dark inner and outer border vs. hood markings absent; (2) throat pale, followed by a broad light brown crossband vs. throat dark, followed by two or three dark broad crossbands; (3) fangs not adapted for spitting, venom discharge orifice large vs. adapted for spitting, discharge orifice smaller.

The new species differs from *Naja philippinensis* by (1) hood markings usually a distinct pale oval marking with narrow dark inner and outer border (vs. without any distinctive markings anteriorly); (2) subcaudal scales 51.4 ± 9.7 (45–61, *n* = 33) (vs. 38–47); (3) distribution confirmed to southwestern China, expected in adjacent Indochina (vs. endemic to Philippines).

The new species differs from *Naja samarensis* by (1) more ventrals 179–205 (195.4 ± 6.7, *n* = 33) vs. 170–179 (174.2 ± 2.4, *n* = 19); (2) dorsal scales near vent usually 15 rows vs. 13 rows; (3) dorsum coloration of adults light brown, regular narrow crossbands on body and tail with dark fringes vs. color above brown to black, usually with a trace of a light lateral line on outer two scales rows; (4) throat pale vs. throat and first few ventrals yellowish; (5) distribution confirmed to southwestern China, expected in adjacent Indochina vs. endemic to Philippines.

The new species differs from *Naja siamensis* by (1) more ventrals 179–205 (195.4 ± 6.7, *n* = 33) vs. 153–174 (? ± ?, *n* = 67); (2) hood markings usually monocellate, a pale oval marking with narrow dark inner and outer border vs. U-, V-, or H-shaped spectacle; (3) dorsum coloration in adults light brown, with single regular buff crossbands with dark fringes on body and tail vs. brightly contrasting black and white pattern in central plain of Thailand, also in Laos; uniformly blackish-brown or black in southeastern and western Thailand.

The new species differs from *Naja sputatrix* by (1) regular narrow crossbands present on body and tail of adults and juveniles with dark fringes vs. never any light dorsal crossbands (except, on rare occasions, a light band behind the hood); (2) throat area pale, lateral spots distinct vs. no clearly defined light throat area, or very dusky and indistinct; lateral spots often missing; (3) adults usually with clear hood markings vs. usually lacking; (4) venom discharge orifice on fang relatively large vs. relatively small.

The new species differs from *Naja sumatrana* by (1) small scales between posterior chin shields usually three (40%) or two (37%), rarely four (13%) or one (10%) (*n* = 30) vs. one (78%) or two (22%), (*n* = 9); (2) dorsum coloration of adults light brown vs. color above black or dark brown; (3) regular single narrow crossbands present on middle and posterior parts of the dorsum (3–15, 7.9 ± 2.7, *n* = 32), and dorsal surface of tail (1–6, 4.2 ± 1.1, *n* = 32) of both adults and juveniles, buff-colored with dark fringes on both edges vs. mostly without light-colored crossbands on body (88%), some juveniles with three to seven narrow light-colored crossbands at posterior body (12%), *n* = 51; (4) a broad light brown crossband present on after throat vs. venter dark or light but without a distinctive black crossband on anterior portion.

**Description of holotype**. Adult male. Head broad, triangular, widest at middle parietals. Body dimensions: snout-vent length 800 mm; tail length 135 mm, about 14% of total length; head length 30.3 mm (snout to end of parietal suture); maximum head width 19.8 mm, about 65% of head length; maximum head height 12.8 mm, about 42% of head length; distance between eyes 14.2 mm, 47% of head length.

**Body scalation**. Ventrals 190, preventrals 2; subcaudals 50, paired, terminating in a spine. Dorsal scales smooth, 20 on neck, 20 at midbody, 15 one head length ahead of vent; elongated, gradually broader toward ventrals; vertebral scales similar to dorsal scales.

**Head scalation**. Scales smooth. Rostral nearly U-shaped, slightly visible in dorsal view. Nasals large, including one prenasal and one postnasal, nostril mainly surrounded by postnasal. External nares moderate, higher than wide, half of vertical diameter of eye. Preocular 1/1, bordering widely with third supralabial, supraocular, prefrontal, and slightly with eye, internasals and postnasals. Loreal scale absent. Internasal paired, widely contact with each other. Prefrontal paired, in shape of right trapezoid, widely contact each other. Frontal moderate, shield-shaped, size similar as prefrontal. Parietals paired, large, largest length about 1.6 times of frontal length, each bordering frontal, supraocular, upper postocular, anterior and posterior temporals, and two small scales on upper side of the head. Eyes rounded; moderate, vertical diameter equals height of fourth supralabial. Postoculars 3/3, similar in size, about half of preocular, the lowest widely in contact with the fourth and fifth supralabial. Temporals 2+3/2+3; lower anterior temporals hexagonal, longer than high; lowest posterior temporal largest, shape irregular. Supralabials 7/7; first two smallest, bordering nasals; third and fourth higher than wide, bordering eye; the seventh longest. Mental triangular in front and ventral view of head. Infralabials 9/9; the first to third in contact with anterior chin shields, the first contact with each other, and anterior edge of anterior chin shields; the fourth contacting both anterior and posterior chin shields; the fifth small and poorly developed, barely seen in ventral view of head, leaving the fourth in broad contact with the sixth; the sixth in contact with posterior chin shields; the seventh to ninth elongated, not contacting chin shields. The two pairs of chin shields form a butterfly shape. A total of three scales separating posterior chin shield, arranged in a “Λ”-shape; first of them much smaller than gulars behind, contacting both anterior chin shields; posterior two scales elongated, broadly contacting each other.

**Coloration in preservative**. Head pattern: Dorsal head uniformly light brown; lateral head gradually paler on supralabials; ventral head uniformly cream white. Dorsal pattern: Dorsum light brown; six buff crossbands present on middle to posterior body, width about length of one dorsal scale, with faint dark fringes, not bifurcate near ventrals. A brown monocellate marking with dark edges and dark oval core present on the hood, not reaching ventrals. Interstitial skin feebly lighter than dorsal scales. Dorsal tail also light brown with four crossbands similar with those on body (Figure 8A). Ventral pattern (Figure 8B): throat pale, ventrals 1st–13th cream white; first ventral crossband covering ventrals 14th–19th, faintly brown, the 19th ventral paler; a small dark patch present on ventrolateral neck at lateral edge of 9th–10th ventral and the first row of dorsal scales; ventrals 20th–24th cream; ventrals 25th and following uniformly pale brown except several ventrals more paler at position opposing crossbands on dorsum; coloration on middle and posterior venter similar with coloration on the first broad crossband; two small dark patches present on skin between lateral edge of eighth and ninth with first row of dorsal scales. Ventral tail uniformly pale.

**Coloration in life**. Similar to coloration in preservative, generally browner.

**Fangs**. Fangs firm, short, not exceed third supralabial; not modified for spitting, venom discharge orifice relatively large and elongated.

**Hemipenes**. (Figure 6C,D). Description based on three adult males (Appendix A). Hemipenis bilobed, covered with dense small spines, spines smaller and sparser at proximal part, larger and denser on distal part; spines on distal part well connected at base, forming calyculate fold. Two pairs of weak lateral folds present on the trunk from sulcate view, nearly identical, covered with dense spines. Lower lobes located at three-fourths of the trunk; upper lobes raised at three-fourths of the hemipenes and reaching the base; a transverse constriction divide the lobes from asulcate view. Two pairs of longitudinal folds present above and below the constriction, feebly raised; longitudinal folds above reaching middle of the lobes and joined near the constriction; longitudinal folds below parallel and short, located at three-fourths of the trunk. Sulcus forked, centripetally spiral to the top, tip of sulcus not visible from sulcate view. Sulcus lip well developed and raised, covered with dense spines. Hemipenes reaching eleventh subcaudal when not extruded, hemipenial retractor muscle reaching 23rd subcaudal. Bifurcation ratio 64–71%, hemipenes length/snout–vent length 3.3–4.1%.

**Variation**. (Figure 9 and Figure 10) Body length for adults 578–1165 mm (894 ± 181 mm, *n* = 16), total length 690–1366 mm (1040 ± 208 mm, *n* = 16). The colorations of dorsum and first ventral crossband of juveniles and hatchlings with body length smaller than 578 mm are generally much darker than in larger specimens (Figure 9A,C,E). Coloration on middle and posterior venter of adults mostly uniformly yellowish-cream (Figure 10B,D), light brown near tail; ventral coloration of juveniles is deep brown or dark (Figure 9F), most individuals or hatchings (76%, *n* = 17) with ambiguous light colored crossbands on middle and posterior ventral surface (Figure 9F). The dorsal head of one female paratype (CIB DL2018081005) is almost pale brown, much lighter than dorsum colorations (Figure 9A and Figure 10A). Hood markings mostly monocellate (76%), some specimens with irregular residual patterns of monocle (21%) (Figure 9C), rarely without any pattern (3%, *n* = 29) (Figure 9C), all specimens checked without dark side spots. Small scales between posterior chin shields usually two (37%) or three (40%), rarely one (10%) or four (13%) (*n* = 30). The first ventral cross band start from 12th–16th (16.0th ± 4.4, *n* = 31) ventral.

**Etymology**. The new species is named after Fuxi (伏羲), one of the human ancestors in Chinese mythologies, usually depicted in cultural relics as half-man and half-snake. The common postures of Fuxi resembles a cobra in reared status; this species is named for the impact that snake had in human culture. The specific nomen is a noun in apposition. To identify this species quickly when dealing with snakebites, we suggest “Brown Banded Cobra” as a common English name, for its unique coloration with a brown body and light crossbands on the body and tail. For the Chinese name, we suggest “西南眼镜蛇” (Xī Nán Yǎn Jìng Shé) as it is distributed in the southwestern part of China.

**Distribution and ecology**. The Brown Banded Cobra is currently known from tropical and southern subtropical areas of southwestern China at elevations between 1000–1400 m. Specimens examined in this study are recorded from Renhe District and Miyi County, Panzhihua Prefecture, southwestern Sichuan Province, and Jiangcheng County, Simao District, Menglian County, Ximeng County, and Pingbian County, Yunan Province. This species was also reported from western Guangxi [30,31,32]. This species is expected to be found in adjacent areas including western Guangxi Zhuang Autonomous Region and southwestern Guizhou Province of China, northeastern Myanmar, northern Laos, northern Thailand, and northwestern Vietnam.

This species was found on gentle slopes of open bush or edges of the forest (Figure 10G,H) during daytime. One individual was kept in captivity for about one year and lived well, being fed on mice and toads. According to Yang and Rao [61], the brown banded cobra has a wide spectrum of food including frogs, snakes, birds, and small mammals. Local people from Yunnan Province reported that this species sometimes sneaks into villages and preys on chicks. Local people from Menglian County observed five to dozens of individuals gathering in an abandoned termite nest during winter in Yunnan Province [61]. The Brown-Banded Cobra is venomous, and easily provoked. This species is the perpetrator that caused the largest number of snakebites in Xishuangbanna, Yunnan Province, based on an analysis of 126 snake bite cases caused from 2007 to 2014 [62].

## 4. Discussion

### 4.1. The Taxonomy of Some Asian Cobras

The population of *Naja naja* (=*N. polyocellata*
**comb. nov.**) from Sri Lanka: *Naja naja,* is widely distributed throughout the India subcontinent and Sri Lanka [22,25]. There are multiple color and pattern variations within the populations in the present geographical distribution of *N. naja*, which resulted in the description of five subspecies by Deraniyagala [8,9,10,11] from the Indian subcontinent, namely *N. n. gangetica* Deraniyagala, 1945 (Gangetic Plain), *N. n. madrasiensis* Deraniyagala, 1945 (Southern India), *N. n. indusi* Deraniyagala, 1960 (Northwestern India, Pakistan), *N. n. bombaya* Deraniyagala, 1961 (Maharashtra of western India), and *N. n. karachiensis* Deraniyagala, 1961 (a black form from southern Pakistan), and one subspecies from Sri Lanka, *N. n*. *polyocellata* Deraniyagala 1939. These subspecies were synonymized with *N. naja* by Wüster [16] and further discussed based upon a morphometric character analysis [14]. However, the insular population of Sri Lanka is different from continental populations by having 15 or more (sometimes up to 20) dark ventral bands vs. one to four bands [8,10,14]. Furthermore, the venom composition of the Sri Lanka population is different from that of the Indian populations in quality and quantity, whereas the antibody binding affinities towards venom of the Sri Lanka population is lower than that of the Indian populations [4]. The Sri Lanka population is divergent from continental populations from Pakistan and Nepal with distinct genetic distance (4.3–4.9% for cyt *b*), which is larger than those between some known specie pairs *N. siamensis* and *N. mandalayensis* (3.2–3.9% for cyt *b*), *N. sumatrana* and *N. mandalayensis* (3.2–3.9% for cyt *b*). These indicates that the subspecies *N. naja polyocellata* Deraniyagala, 1939 should be resurrected and recognized as a full species, *N. polyocellata*
**comb. nov.** Deraniyagala, 1939. Further taxonomic research that combines molecular, morphological and ecological methods is needed for the *N. naja* complex.

The populations of *Naja sumatrana* (=*N. s. miolepis*) from Borneo and Palawan Islands: These populations were described as a subspecies *Naia tripudians miolepis* Boulenger, 1896 based on their juveniles having a unique “V”-shaped hood pattern [19]. It was synonymized with *N. sumatrana* by Wüster [15] based on a morphometric character analysis. The *N. sumatrana* sample from the Philippines (distributed in Palawan Islands) clustered into a clade with one sample from Malaysia (Figure 2). Additionally, it is divergent from two samples from Indonesia and Malaysia with genetic distance (2.9–3.4% for cyt *b*). These support the resurrection of the subspecies *N. sumatrana miolepis* Boulenger, 1896. However, further studies on adult morphological comparisons and distribution boundary of *N. sumatrana sumatrana* and *N. sumatrana miolepis* are still needed to determine whether they are different species.

The population of southeastern Asian *Naja kaouthia* (= *N. naja isanensis*?): A cream-colored subspecies, *N. kaouthia suphanensis*, Nutaphand,1986 was described from central Thailand [63]. The subspecies was then synonymized with *N. kaouthia* by Wüster [13] based on a multivariate morphometric analysis between *N. k. suphanensis* and *N. kaouthia* from central Thailand, but no specimens from or near the type locality of *N. kaouthia*, the Bengal region [57], were included in the analysis. The southeastern Asian *N. kaouthia* population from Myanmar, Thailand, and Vietnam differs from *N. kaouthia* from the lowlands in Bengal and adjacent areas by usually no crossbands on body and a few (usually one or two) clear crossbands on neck after the hood marking (21%) (*n* = 24) vs. parallel light-colored crossband pairs or densely woven lines on body and tail, or at least with a clear crossband on the neck behind the hood markings (*n* = 24)). However, the coloration of southeastern Asian population is similar to populations of *N. kaouthia* from the southern slope of the Himalayan Mountains and the Southeast Asian *N. kaouthia* population do not spit venom (based on the observation of more than 100 individuals), whereas multiple cases (*n* > 16) were recorded in populations from eastern India and Nepal [59,64]. A cryptic species was proposed from the northeastern population of *N. kaouthia* by Ratnarathorn et al. [5]. This population is the phylogenetic sister to *N. atra* (“*N. naja*”) and *N. kaouthia* from Thailand based on the Control Region with weak support (BPP 0.65) (Figure S1 of [5]). However, the northeastern population is sister to *N. siamensis* based on partial cyt *b* (about 603 bp) with much stronger support (BPP 0.986) (Figure S2 of [5]). Due to only four operational taxonomic units (OTUs) (*N. atra*, *N. siamensis*, the northeastern population of *N. kaouthia*, and *N. kaouthia*) were included in the analysis, the phylogenetic position of the cryptic species is still doubtful. A brown spitting population of cobra with indistinct spectacle mark from northeastern Thailand has been described as *N. naja isanensis* Nutaphand, 1982 [65] and it was treated as a color variety of *N. siamensis* by Wüster and Thorpe [66]. This population is possibly phylogenetically the sister to *N. siamensis*, and morphologically resembles “*N. kaouthia*” from Thailand due to the brown coloration and indistinct spectacle mark, so, it is possible the *nomen Naja naja isanensis* represents the northeastern population of *N. kaouthia* in Ratnarathorn et al. [5]. Since the samples of *N. kaouthia* from Myanmar to Vietnam form a clade (A; 98/1.00) sister to insular species *N. sagittifera* and have minor genetic divergence within clade (0.0% to 1.2% for cyt *b*), these samples should represent the same species or subspecies. However, the name for this taxon remains unsolved; the status of subspecies *N. k. suphanensis* and *N. n. isanensis* needs to be further studied.

### 4.2. Geographical Variations of N. kaouthia

The coloration form of *N. kaouthia* varies among geographic regions. Those specimens from the southern slope of the southeastern Himalayas (Sikkim, Bhutan and southern Tibet, China) have typically monocellate hood markings and uniformly brown dorsum coloration. However, specimens from West Bengal and adjacent Odisha, India demonstrated clear crossbands of various forms. Similar situation is also found in *N. atra*, no mitochondrial gene differentiation was apparent among different ventral color morphs, the color morph is more related to the geographic populations [33]. The venom composition of *N. kaouthia* varies between the southern Himalayan population and lowland population from West Bengal [6]. These geographic variations might be related to ecological differences. To clarify these questions, biogeographic research for *N. kaouthia* based on comprehensive sampling in these areas is highly recommended to build a strong framework for snakebite medical treatment and antivenin development.

### 4.3. Spitting Behavior and Venom Discharge Orifice

The venom discharge orifice on fangs of *N. atra*, *N. kaouthia* and *N. fuxi* are relatively large and elongated. *N. atra* and *N. kaouthia* were listed as non-spitting species according to Wüster and Thorpe [16] and Young et al. [50]. However, there are cases reported for *N. kaouthia* spitting venom in eastern India and Nepal [59,64]. There is also a case for *N. atra* from southern China spitting venom with description for relative venom discharge orifice [56]. According to personal observation on *N. **fuxi***
**sp. nov.** by Liang Zhang, the cobra could spit venom as far as about 0.6 m. The venom discharge orifices of the Asian cobras in Clade I (except *N. sumatrana miolepis*, unknown) are small, while those for species in clade D are relatively larger [17,56,67] (this study). The Asian spitter clade J originated ~2.5 Ma, which is contemporaneous with the beginning of the quaternary glaciation [51,68]. The less spitting-adapted species of clade D with larger venom discharge orifices, also originated ~2.5 Ma (Figure S14 of [51]). This suggests that venom spiting behavior evolved independently twice in subgenus *Naja* and the latter one (clade D) might be relevant to climate change and distribution change of *H. erectus* during the quaternary glaciation.

## 5. Conclusions

Our results led to taxonomic revisions of some Asian cobras: (1) the former Chinese population of *N. kaouthia* represents a new species, *N. fuxi*
**sp. nov.**; (2) the subspecies *N. naja polyocellata* was resurrected and recognized as a full species *N. polyocellata*
**comb. nov.**; (3) the subspecies *N. sumatrana miolepis* was resurrected. This study highlights the necessity to evaluate effectiveness of cobra antivenin based on comprehensive taxonomic frameworks. However, there are still some open questions about the systematics of widespread species such as *N. kaouthia*. The taxonomy of Asian cobras still needs further revision. A wide international collaboration network is recommended to construct a solid taxonomic framework of these medically important species for the common well-being of Asian people.

## Figures and Tables

**Figure 1 animals-12-03481-f001:**
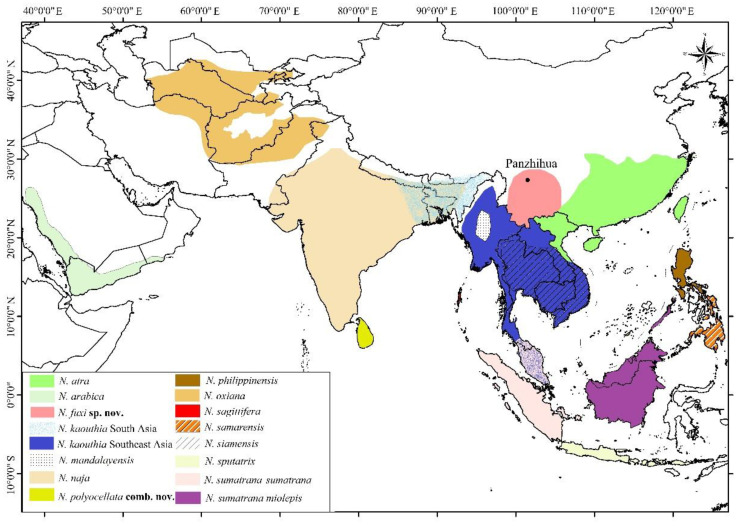
Distributions of cobras in Asia. Distribution data were adjusted from IUCN https://www.iucnredlist.org (accessed on 26 October 2022).

**Figure 2 animals-12-03481-f002:**
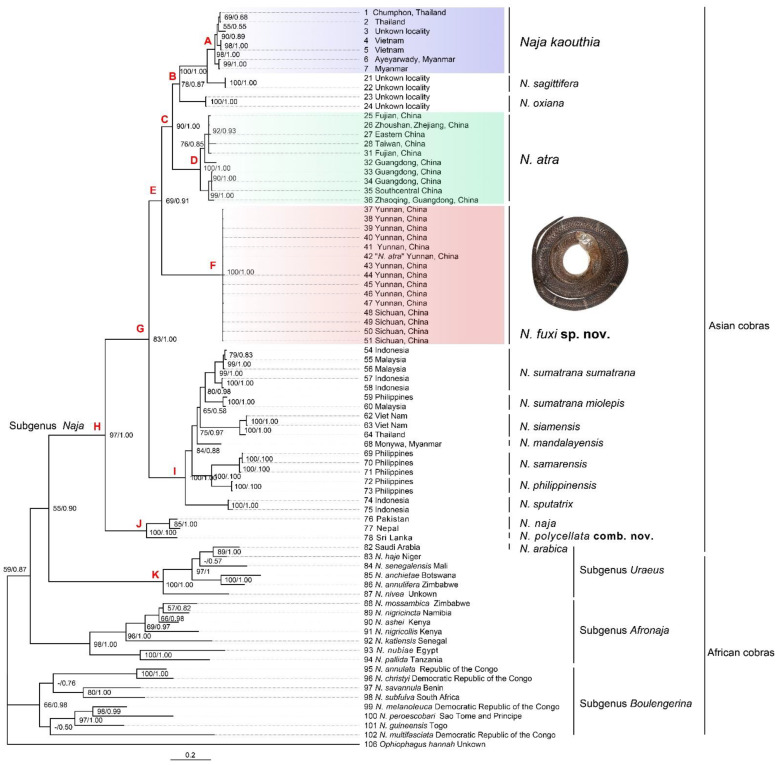
Phylogenetic trees of the genus *Naja* inferred by Bayesian analyses (BI) based on concatenated mitochondrial gene alignment for 1028 bp cyt *b* and 659 bp ND4. Both bootstrap supports (BS) and Bayesian posterior probabilities (BPP) are indicated on each of the corresponding node. Support values for weekly supported (BS < 50, BPP < 0.50) nodes are indicated as “-”.

**Figure 3 animals-12-03481-f003:**
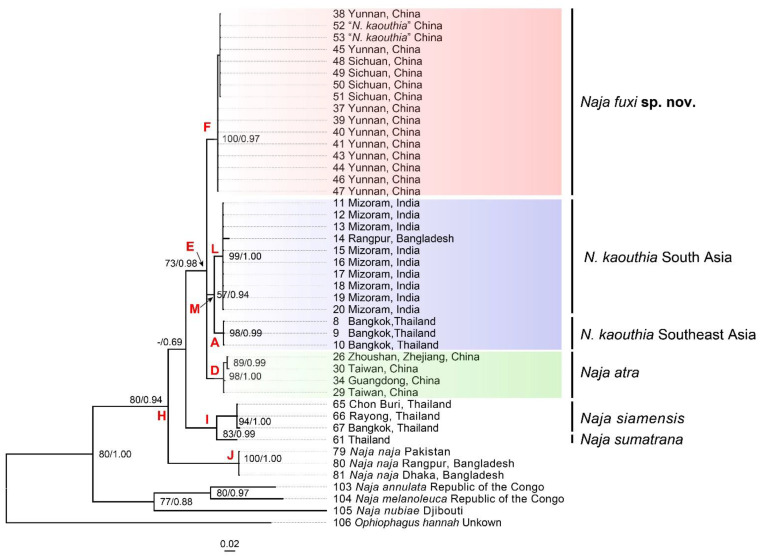
Phylogenetic trees of the genus *Naja* inferred by Bayesian analyses (BI) based on 627 bp mitochondrial gene COI. Both bootstrap supports (BS) and Bayesian posterior probabilities (BPP) are indicated on each of the corresponding node. Support values for weekly supported (BS < 50, BPP < 0.50) nodes are indicated as “-”.

**Figure 4 animals-12-03481-f004:**
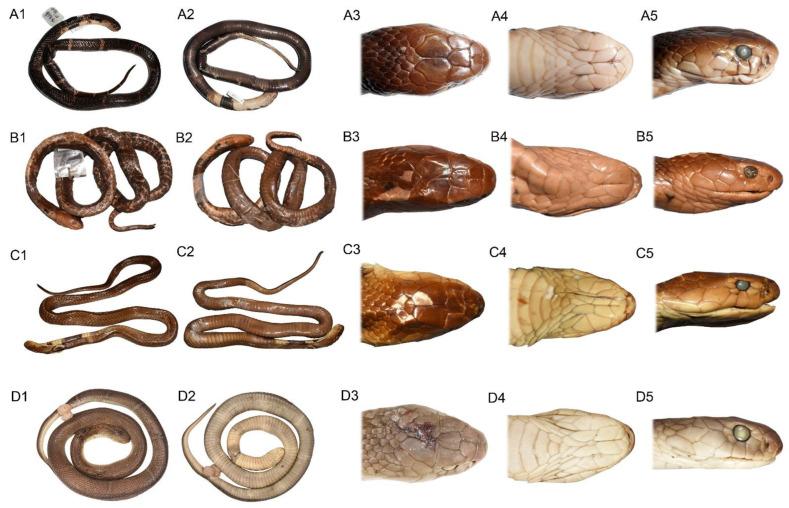
Body and head morphological comparisons between *Naja atra* (**A**), *N. kaouthia* South Asian clade (**B**), *N. kaouthia* Southeastern Asian clade (**C**) and *N. fuxi*
**sp. nov.** (**D**). (**A**) subadult female (body length 730 mm) neotype of *N. atra* CIB12273 from Zhoushan Island, Zhejiang Province; (**B**) adult female *N. kaouthia* ZMH R04803 from Port Canning, West Bengal, India; (**C**) adult male ZMH R02885 of *N. kaouthia* from Bangkok, Thailand; (**D**) adult male holotype of *N. fuxi*
**sp. nov.** CIB DL2018053147 from Panzhihua, Sichuan, China. 1: dorsal body; 2: ventral body; 3: dorsal head; 4: ventral head; 5: lateral head. (**A**,**D**) photographed by Sheng-Chao Shi; (**B**,**C**) by Gernot Vogel.

**Figure 5 animals-12-03481-f005:**
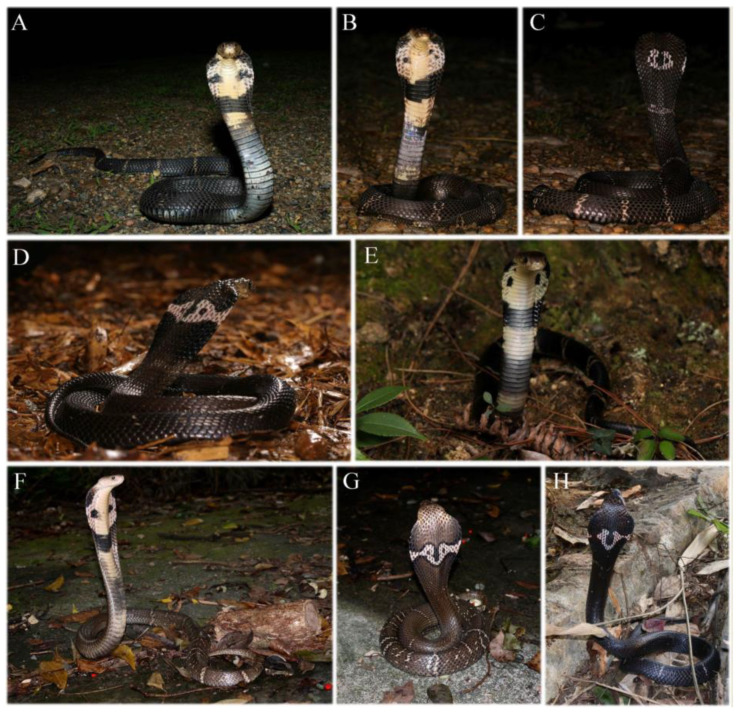
*Naja atra* in life. (**A**) Adult from Qingtian, Zhejiang, China; (**B**–**D**) one adult (**B**,**C**) and another (**D**) from Lishui, Zhejiang, China; (**E**) juvenile from Wuyi, Fujian, China; (**F**,**G**) one adult from Guangzhou, Guangdong, China; (**H**) adult from Guangzhou, Guangdong, China. (**A**–**E**) Photographed by Bin-Qing Zhu; (**F**–**H**) photographed by Liang Zhang.

**Figure 6 animals-12-03481-f006:**
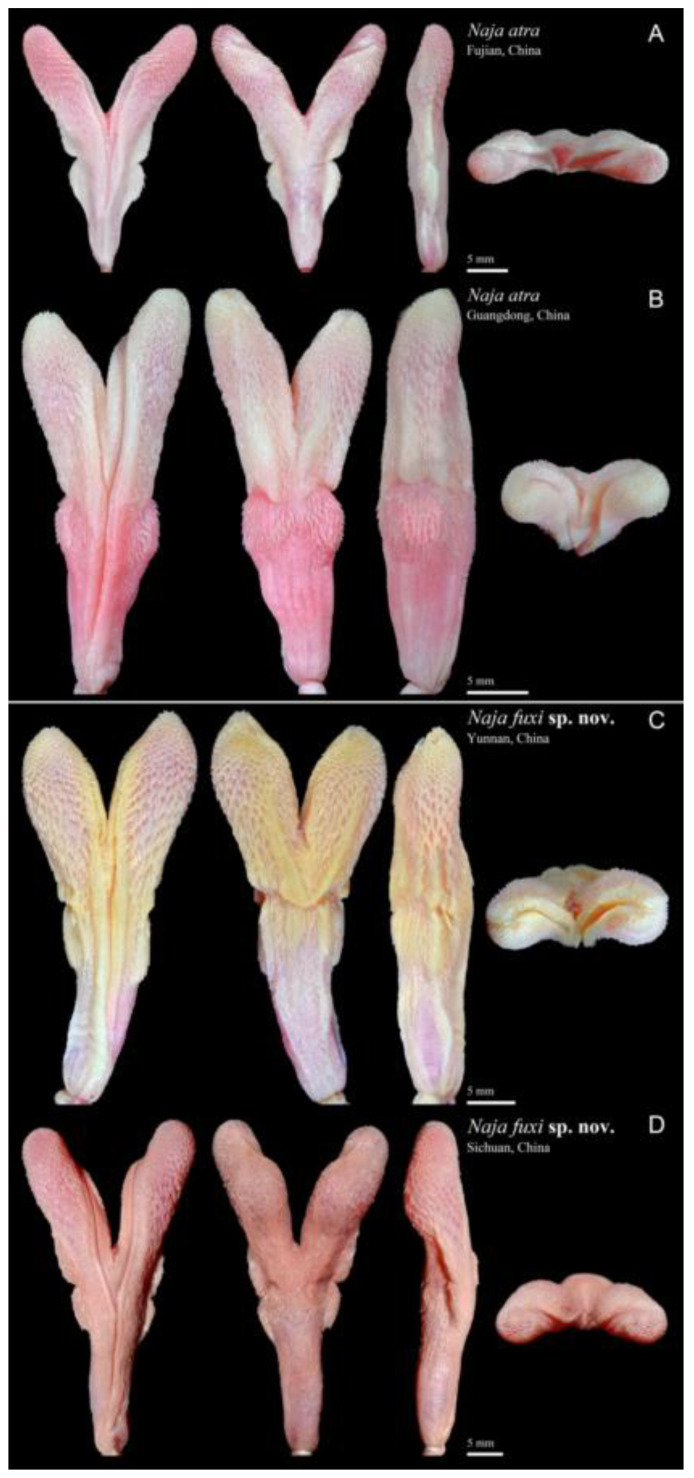
Hemipenis of *Naja atra* ((**A**) CIB LJT-FJ22020060; (**B**) CIB LJT-GD2020103) and *N. fuxi*
**sp. nov.** ((**C**) KIZ 030071; (**D**) CIB DL2018053147). Sulcate, asulcate, lateral and top view from left to right. Photographed by Jun-Jie Huang and Jin-Long Ren.

**Figure 7 animals-12-03481-f007:**
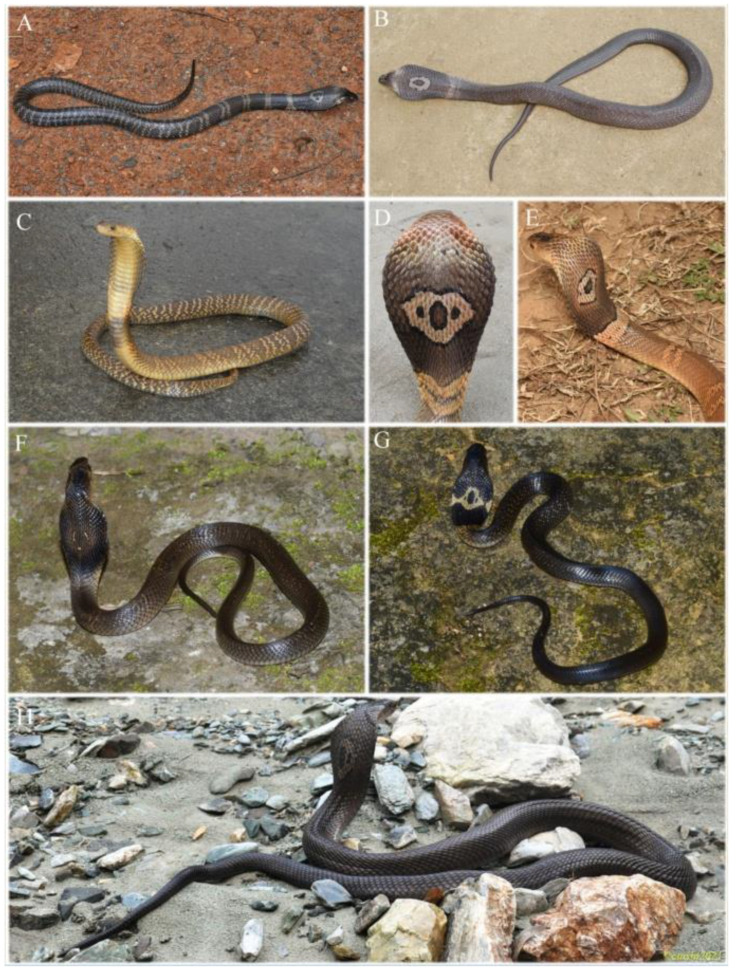
South Asian *Naja kaouthia* in life. (**A**,**B**) Two adults from Jhargram, West Bengal, India; (**C**) Individual from Kolkata outskirts of West Bengal, India; (**D**) Adult from Siliguri, North Bengal region of West Bengal, India; (**E**) Individual from Bhuvneshwar, Odisha, India; (**F**,**G**) two juveniles from Mizoram, India; (**H**) Individual from Samtse town, Bhutan. ((**A**–**E**) photographed by Vivek R Sharma; (**F**,**G**) photographed by Gernot Vogel; (**H**) photographed by Choggyal Taz).

**Figure 8 animals-12-03481-f008:**
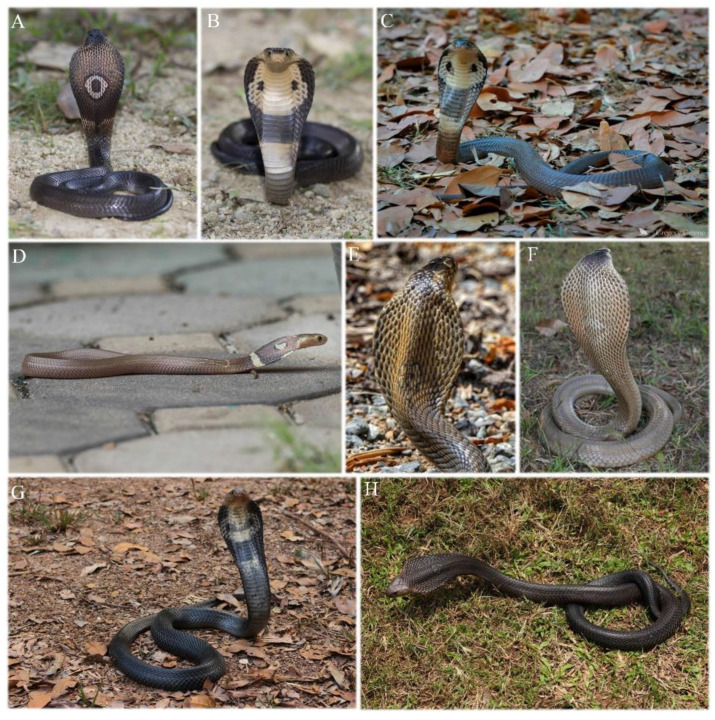
Southeastern Asian *Naja kaouthia* in life. (**A**,**B**) a juvenile from Selangor, Peninsular Malaysia; (**C**) an adult from Chang Wat Nakhon Ratchasima, Thailand; (**D**) a juvenile from Samut Prakan, Thailand; (**E**) an adult from Surat Thani, Thailand; (**F**) an adult from Ranong, Thailand; (**G**,**H**) two adults from Duc Trong, Lam Dong, Vietnam. (**A**) was photographed by Dr. Teo Eng Wah. Other photos were cited from iNaturalist.org occurrence dataset https://doi.org/10.15468/ab3s5x via GBIF.org (licensed under http://creativecommons.org/licenses/by-nc/4.0/) by following photographer: Gregory Greene ((**C**) record No. 2006052432), Wich’yanan L ((**D**) record No. 3090707753), Mintkhaosok ((**E**) record No. 3325726349), Knotsnake ((**F**) record No. 3384192379), Herpingvietnam ((**G**) record No. 2557801804), Leonid A. Neymark ((**H**) record No. 2366151765).

**Figure 9 animals-12-03481-f009:**
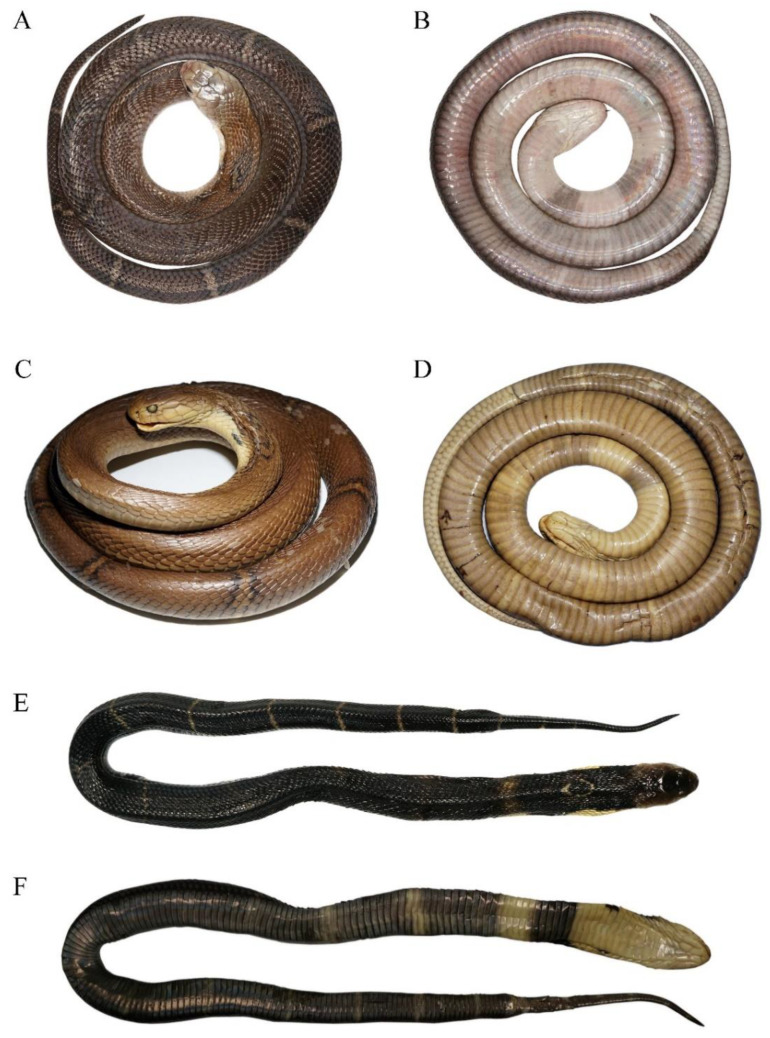
Paratypes of *Naja fuxi*
**sp. nov.** (**A**,**B**) Dorsal and ventral view of adult female paratype CIB DL2018081005 from type locality Panzhihua, Sichuan before preservation; (**C**,**D**) Dorsolateral and ventral view of adult male KIZ030071 from Simao, Pu’er, Yunnan in preservative; (**E**,**F**) Dorsal and ventral view of juvenile CIB YNJC0068 from Jiangcheng, Yunnan, China before preservation. (Photographed by Sheng-Chao Shi).

**Figure 10 animals-12-03481-f010:**
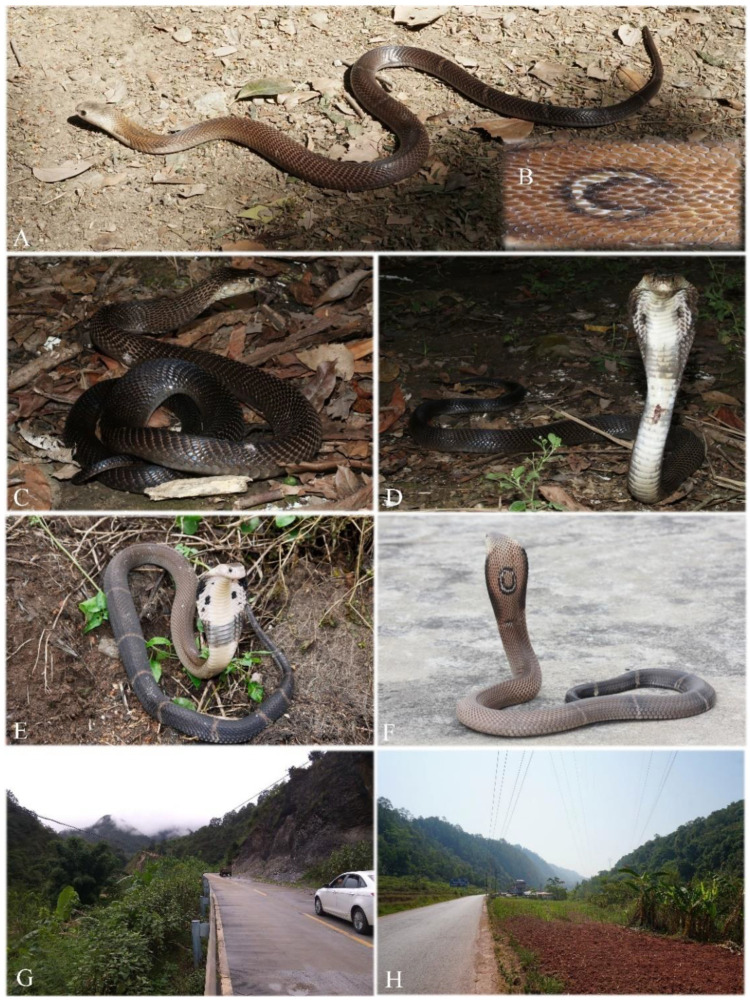
*Naja fuxi***sp. nov.** in life and habitats. (**A**,**B**) Dorsolateral view and hood pattern of adult female paratype CIB DL2018081005 from Panzhihua, Sichuan, China; (**C**,**D**) Dorsolateral and front view of an unvouchered adult from Jiangcheng, Pu’er, Yunnan, China; (**E**,**F**), two adults from Wenshan, Yunnan, China; (**G**) Subtropical Forest at type locality Panzhihua, Sichuan, China; (**H**) Tropical monsoon forest and farmland at Jiangcheng, Pu’er, Yunnan, China. (Photos (**A**,**B**,**H**) by Sheng-Chao Shi; (**C**,**D**,**G**) by Li Ding; (**E**,**F**) by Liang Zhang).

**Table 1 animals-12-03481-t001:** Samples analyzed in this study.

Sample ID	Taxa	Voucher Numbers	Locality	cyt *b*	ND4	COI
1	*Naja kaouthia*	WW 585	Chumphon Province, Thailand	GQ359507	EU624209	
2	*N. kaouthia*	isolate 818	Thailand	MT346707	MT346903	
3	*N. kaouthia*		Unknown	LR880477		
4	*N. kaouthia*	isolate 813	Vietnam	MT346706	MT346902	
5	*N. kaouthia*	isolate 812	Vietnam	MT346705	MT346901	
6	*N. kaouthia*	CAS 206602	Ayeyarwady, Myanmar	AF217835	AY058982	
7	*N. kaouthia*	isolate 839	Myanmar	MT346708	MT346904	
8	*N. kaouthia*	NKA3	Bangkok, Thailand			AB920185
9	*N. kaouthia*	NKA2	Bangkok, Thailand			AB920184
10	*N. kaouthia*	NKA1	Bangkok, Thailand			AB920183
11	*N. kaouthia*	PUCZM/X/SL787	Mizoram, India			MH107858
12	*N. kaouthia*	PUCZM/X/SL774	Mizoram, India			MH107857
13	*N. kaouthia*	PUCZM/X/SL773	Mizoram, India			MH107856
14	*N. kaouthia*	CUZSI81	Badarganj, Rangpur, Bangladesh			KM521202
15	*N. kaouthia*	MZMU1163	Aizawl, Mizoram, India			MT348385
16	*N. kaouthia*	MZMU982	Mizoram University Campus, India			MT348386
17	*N. kaouthia*	MZMU1170	Aizawl, Mizoram, India			MT348387
18	*N. kaouthia*	MZMU998	Mizoram University Campus, India			MT348388
19	*N. kaouthia*	MZMU1426	Mizoram University Campus, India			MT348389
20	*N. kaouthia*	MZMU1195	Mizoram University Campus, India			MT348390
21	*N. sagittifera*	isolate 400	Unknown	MT346720	MT346916	
22	*N. sagittifera*	isolate 1815	Unknown	MT346721	MT346917	
23	*N. oxiana*	isolate 838	Unknown	MT346714	MT346910	
24	*N. oxiana*	isolate 832	Unknown	MT346713	MT346909	
25	*N. atra*	LJT-FJ2020060	Dongping Mountain, Xiamen, Fujian, China	ON221325	ON221392	
26	*N. atra*	GXNU 2021070801	Zhoushan Island, Zhejiang, China	ON221326		
27	*N. atra*	C29	Eastern China	JN160670		
28	*N. atra*	isolate 793	Taiwan, China	MT346704	MT346900	
29	*N. atra*	RN1570	Taiwan, China			KP749826
30	*N. atra*	RN1298	Taiwan, China			KP749825
31	*N. atra*	CIB 20190430	Wuyishan, Fujian, China	ON221327		
32	*N. atra*	CIB 2021042007	Luofushan, Guangdong, China	ON221328		
33	*N. atra*	CIB CR430	Guangzhou, Guangdong, China	ON221329	ON237986	ON221393
34	*N. atra*	CIB 093930	Purchased from Guangzhou, Guangdong Province, China	EU913475	EU913475	EU913475
35	*N. atra*	C17	Southcentral China	JN160658		
36	*N. atra*	LJT-GD2020103	Dinghu Mountain, Zhaoqing, Guangdong, China	ON221330		
37	*N. fuxi* **sp. nov.**	KIZ 090071	Simao, Pu’er, Yunnan, China	ON221331		ON237987
38	*N. fuxi* **sp. nov.**	KIZ F20180066	Simao, Pu’er, Yunnan, China	ON221332		ON237988
39	*N. fuxi* **sp. nov.**	KIZ 20180801	Simao, Pu’er, Yunnan, China	ON221333		ON237989
40	*N. fuxi* **sp. nov.**	KIZ 2020090301	Menglian, Pu’er, Yunnan, China	ON221334	ON221394	ON237990
41	*N. fuxi* **sp. nov.**	KIZ 2020091201	Ximeng, Pu’er, Yunnan, China	ON221335	ON221395	ON237991
42	*N. fuxi* **sp. nov.** (“*N. atra*”)	C31	Pingbian, Yunnan, China, China	JN160672		
43	*N. fuxi* **sp. nov.**	CIB YNJC0022	Jiangcheng, Pu’er, Yunnan, China	ON221336	ON221396	ON237992
44	*N. fuxi* **sp. nov.**	CIB DL000020	Jiangcheng, Pu’er, Yunnan, China	ON221337	ON221397	ON237993
45	*N. fuxi* **sp. nov.**	CIB DL000070	Jiangcheng, Pu’er, Yunnan, China	ON221338	ON221398	ON237994
46	*N. fuxi* **sp. nov.**	CIB DL000096	Jiangcheng, Pu’er, Yunnan, China	ON221339		ON237995
47	*N. fuxi* **sp. nov.**	CIB DL000249	Jiangcheng, Pu’er, Yunnan, China	ON221340	ON221399	ON237996
48	*N. fuxi* **sp. nov.**	CIB 2018053147	Panchizhua, Sichuan, China	ON221341		ON237997
49	*N. fuxi* **sp. nov.**	CIB 098874	Miyi, Panzhihua, Sichuan, China	ON221342		ON237998
50	*N. fuxi* **sp. nov.**	CIB 098875	Miyi, Panzhihua, Sichuan, China	ON221343		ON237999
51	*N. fuxi* **sp. nov.**	CIB YS2	Yanyuan, Panzhihua, Sichuan, China	ON221344	ON221400	ON238000
52	*N. fuxi* **sp. nov.** (“*N. kaouthia*”)	CHS026	China			MK064598
53	*N. fuxi* **sp. nov.** (“*N. kaouthia*”)	CHS726	China			MK064840
54	*N. sumatrana sumatrana*	isolate 589	Indonesia	MT346737	MT346933	
55	*N. sumatrana sumatrana*	isolate 587	Malaysia	MT346736	MT346932	
56	*N. sumatrana sumatrana*	isolate 586	Malaysia	MT346735	MT346931	
57	*N. sumatrana sumatrana*	isolate 295	Indonesia	MT346734	MT346930	
58	*N. sumatrana sumatrana*	isolate 294	Indonesia	MT346733	MT346929	
59	*N. sumatrana miolepis*	isolate 1827	Philippines	MT346738	MT346934	
60	*N. sumatrana miolepis*	isolate 188	Malaysia	MT346732	MT346928	
61	*N. sumatrana*	NSU1	Thailand			AB920186
62	*N. siamensis*	isolate 811	Viet Nam	MT346728	MT346926	
63	*N. siamensis*	isolate 810	Viet Nam	MT346727	MT346925	
64	*N. siamensis*	isolate 26	Thailand	MT346725	MT346923	
65	*N. siamensis*	SC01f	Bangkok, Thailand			LC086063
66	*N. siamensis*	NSI3	Rayong, Thailand			AB920188
67	*N. siamensis*	NSI2	Bangkok, Thailand			AB920187
68	*N. mandalayensis*	CAS 204375-6	Monywa, Myanmar	AF155211		
69	*N. samarensis*	isolate 1803	Philippines	MT346723	MT346919	
70	*N. samarensis*	isolate 1806	Philippines	MT346724	MT346920	
71	*N. samarensis*	isolate 841	Philippines	MT346722	MT346918	
72	*N. philippinensis*	isolate 1828	Philippines	MT346719	MT346915	
73	*N. philippinensis*	isolate 1825	Philippines	MT346718	MT346914	
74	*N. sputatrix*	isolate 584	Indonesia	MT346730	MT346922	
75	*N. sputatrix*	isolate 583	Indonesia	MT346729	MT346921	
76	*N. naja*	ZMUVAS8	Pakistan			MK936173
77	*N. naja*	isolate 579	Nepal	MT346711	MT346907	
78	*N. polyocellata* **comb. nov.**	isolate 581	Sri Lanka	MT346712	MT346908	
79	*N. naja*	ZMUVAS21	Pakistan			MK941841
80	*N. naja*	BDS80/CUZSI80	Pirganj, Rangpur, Bangladesh			KM521201
81	*N. naja*	DUZM_S052.1	Dhaka, Bangladesh			MT215095
82	*N. arabica*	isolate 1678/WW 1678	Taif, Saudi Arabia	GQ387104	GQ387075	
83	*N. haje*	isolate 1653	northern Nigeria	GQ387099	GQ387070	
84	*N. senegalensis*	isolate 2018	Mali	GQ387112	GQ387083	
85	*N. anchietae*	isolate 1892	Botswana	MT346741	GQ387087	
86	*N. annulifera*	isolate 881	Zimbabwe	GQ359504	GQ359586	
87	*N. nivea*	isolate 1482	Unknown	MT346760	MT346936	
88	*N. mossambica*	isolate 882	Zimbabwe	MT346655	MT346848	
89	*N. nigricincta*	isolate 877	Namibia	MT346661	MT346854	
90	*N. ashei*	isolate 1394	Kenya	MT346647	MT346842	
91	*N. nigricollis*	isolate 3103	Kenya	MT346680	MT346874	
92	*N. katiensis*	isolate 2022	Senegal	MT346653	MT346845	
93	*N. nubiae*	isolate 837	Egypt	GQ359497	GQ359579	
94	*N. pallida*	isolate 1080	Tanzania	GQ359496	GQ359578	
95	*N. annulata*	isolate 2717	Republic of the Congo	MT346700	MT346896	
96	*N. christyi*	isolate PM147	Democratic Republic of the Congo	MT346701	MT346897	
97	*N. savannula*	isolate 2046	Benin	MH337600	MH337406	
98	*N. subfulva*	isolate 2654	South Africa	MH337633	MH337439	
99	*N. melanoleuca*	isolate 2721	Democratic Republic of the Congo	MH337589	MH337395	
100	*N. peroescobari*	isolate 1197	Sao Tome and Principe	MH337634	MH337440	
101	*N. guineensis*	isolate 2491	Togo	MH337580	MH337386	
102	*N. multifasciata*	isolate 4313	Democratic Republic of the Congo	MT346702	MT346898	
103	*N. annulata*	UPRP:558270	Impongui, Likouala, Republic of the Congo			MH274482
104	*N. melanoleuca*	UPRP:558271	Impongui, Likouala, Republic of the Congo			MH274485
105	*N. nubiae*	USNM:Herp:589595	Tadjourah, Day (village), Djibouti			MG700028
106	*Ophiophagus hannah*	CIB093929	Unkown	EU921899	EU921899	

**Table 2 animals-12-03481-t002:** Uncorrected *p*-distances based on cyt *b* genes for subgenus *Naja.* (Values in %).

No.	Species (Sample Numbers in Table 1)	1	2	3	4	5	6	7	8	9	10	11	12	13
1	*Naja kaouthia* Southeast Asia (Myanmar; Thailand; Vietnam) (1–7)	0.0–1.2												
2	*N. sagittifera* presumed Andaman Islands (21–22)	**2.4–2.9**	0.0											
3	*N. oxiana* Unkown locality (23–24)	5.0–5.5	5.3–5.4	0.0										
4	*N. atra* Southern and Southeastern China (25–28, 31–36)	**4.6–6.3**	6.1–6.7	5.2–5.8	0.1–2.1									
5	*N. fuxi***sp. nov.** Southwestern China (Sichuan and Yunnan) (37–51)	**5.3–6.7**	6.2–6.4	**4.3–4.4**	**4.1–5.0**	0.0–0.3								
6	*N. sumatrana sumatrana* Indonesia and Malaysia (54–58)	8.2–9.2	8.5–8.9	7.4–7.8	6.9–7.6	6.7–7.5	0.0–1.2							
7	*N. sumatrana miolepis* (Malaysia and Philippines (59–60)	8.4–9.1	7.8–8.3	7.5	7.6–8.4	7.6–7.8	**2.9–3.4**	0.2						
8	*N. siamensis* Viet Nam and Thailand (62–64)	9.7–0.9	9.1–9.7	8.4–9.1	8.8–10.1	8.1–8.9	**4.3–5.0**	**4.7–5.0**	0.0–1.7					
9	*N. mandalayensis* Monywa, Myanmar (68)	8.7–9.0	8.4	7.7	8.0–8.8	7.4	**3.2–3.5**	**3.9**	5.0–5.4	/				
10	*N. samarensis* Philippines (68–71)	9.7–10.4	9.4–9.5	8.4	8.8–9.7	8.1–8.5	5.3–6.0	5.8–6.0	7.1–7.6	5.7	0.0–0.3			
11	*N. philippinensis* Philippines (72–73)	9.7–10.1	9.4–9.5	8.4	9.0–9.4	8.1–8.2	5.0–5.4	4.7–5.0	6.5–6.8	5.4	**4.3**	0.0		
12	*N. sputatrix* Indonesia (74–75)	9.2–10.0	8.6–8.8	8.3–8.5	7.7–8.5	7.6–8.2	4.3–4.8	5.4–5.7	6.2–6.6	4.9–5.4	6.3–6.9	6.2–6.5	0.5	
13	*N. naja* Pakistan and Nepal (76–77)	9.5–11.5	9.2–10.9	7.9–9.4	8.7–10.4	9.8–10.4	9.2–107	9.8–107	10.0–11.3	10.5–10.7	10.6–11.5	11.4–11.8	8.1–9.9	**1.4**
14	*N. polyocellata***comb. nov.** Sri Lanka (78)	10.7–11.3	10.4	8.7	9.5–10.4	9.1–9.3	9.4–9.8	9.6–9.8	10.0–10.6	10.0	10.8–11.1	11.7	9.7–10.2	**4.3–4.9**

**Table 3 animals-12-03481-t003:** Uncorrected P-distance of subgenus *Naja* based on COI genes. (Values in %).

No.	Species (Sample Numbers in Table 1)	1	2	3	4	5	6	7
1	*Naja kaouthia* Southeast Asia (Bangkok, Thailand) (8–10)	0.0						
2	*N. kaouthia* South Asia (Mizoram, India and Rangpur, Bangladesh) (11–20)	**1.8–2.4**	0.0–0.6					
3	*N. atra* southeastern China (Zhejiang, Taiwan, Guangdong) (26, 29, 30, 34)	**2.7–3.4**	**3.1–4.2**	0.2–0.6				
4	*N. fuxi***sp. nov.** Southwestern China (Sichuan and Yunnan) (37–41, 43–53)	**2.5–2.9**	**2.0–3.0**	**2.5–3.2**	0.0–0.2			
5	*N. sumatrana* Thailand (61)	6.5	6.2–6.9	6.1–6.3	6.0–6.4			
6	*N. siamensis* Thailand (Chon Buri, Rayong, Bangkok) (65–67)	5.6	5.6–6.2	5.7–6.5	5.4–5.8	**3.7–4.0**	0.0–0.3	
7	*N. naja* Pakistan and Bangladesh (79–81)	7.7	8.1–8.9	7.5–8.2	7.4–8.0	8.5	8.6–8.7	**0.0**

## Data Availability

The data presented in this study are available on request from corresponding author.

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
