# Peer review of "Description of a New Cobra (Naja Laurenti, 1768; Squamata, Elapidae) from China with Designation of a Neotype for Naja atra"

_animals, 2022, doi:10.3390/ani12243481_

Round 1

Reviewer 1 Report

I suggested a few changes regarding the summary statistics and the use of parentheses with "vs."  The two clauses can be correctly used without placing the "vs." data in parentheses, and since you have so many other data enclosed thusly, it makes for easier reading (I think).  If possible (with publisher) I suggest to include the mean symbol (x) when the range of values are not given, just for clarity (my Mac does not have the correct mean symbol with horizontal bar above the x).  I could not comment on the molecular methods and results but I am sure they are fine.  I suggest that the Deraniyagala names have his name and year included after each subspecies (not done on MS as my papers are packed away).

Author Response

  Thank you for your efforts and providing many carefully changes to our manuscript. We deeply appreciate it. We revised the MS according to your detailed suggestion in the pdf file you provided. Changes are shown in the revised MS word file in “track” mode.

  1. We removed those parentheses or “vs.” as you suggested in pdf file you provided.
  2. We added range and the mean symbol  no longer needed.
  3. The name and year of subspecies by Deraniyagala were added.

   Besides, the first diagnose character was rewritten to make it more understandable.

Reviewer 2 Report

The ms revised is devoted to the problem of taxonomic diversity and cryptic diversity among important group of snakes, genus Naja. The text is full enough, methods and materials correspond the goal and tasks of this study. The only comments concern genetic distances. The authors used the value as decimals, but the more tradional mode is to present values in %. It will be desirable to provide more discussion and argue the values of genetic distances between different species. In text they use as "genetic" as well as "genetical" distiances.   

Resuming I recomment to accept this manuscript for publication in Animals after minor revision.  

Author Response

Thank you for your kind comments and suggestions.

  1. Values of genetic distances were adjusted those in %.
  2. Discussion and argue on the genetic distance between species were added in the Results 3.1.
  3. Those "genetical" distances in the text were replaced with "genetic"

Reviewer 3 Report

Dear Authors,

I find your paper interesting. I have only minor remarks listed below.

Methodological remarks:

My only suggestion for this section is to provide Head Length measured as the distance from the snout ti to the posterior edge of quadrate-mandibular articulation in addition to head length measured to the end of parietal suture. This is not obligatory, however, some authors in their systematic, ecological or evlutionaty studies used one of them. Inclusion of this measurement (if possible), let other researchers to refer to this paper.

Results.

Please provide (as supplementary material or in-text table?) summary of morphological measurements of your samples, with both sexes and juveniles listed separately. If this is included in the Supplementary material S3 ignore this comment - unfortunatelly I don't have acess to supplementary materials in the review papers. 

Author Response

  Thank you for your suggestions. Your suggestions are very thoughtful.

  1. Methodological remarks.

Head Length measured from snout tip to the end of parietals suture is a popular accepted concept. We did not measure the Head Length from the snout tip to the posterior edge of quadrate-mandibular articulation. For the moment, we could not access to all of our specimens, sorry that we could not provide this.

  1.  Result

Thank you, it is a good suggestion. We did not provide summary of morphological measurements in different gender, because that genders of some samples are not clear (especially for those samples only have living photos). To make comparisons based on more samples, we try to rely on characters that different among species rather than gender.  
